# Multi-Agent Debate for LLM Judges with Adaptive Stability Detection

**Tianyu Hu**
tyrionhuu@gmail.com

**Zhen Tan**
Arizona State University
ztan36@asu.edu

**Song Wang**
University of Central Florida
song.wang@ucf.edu

**Huaizhi Qu**
UNC Chapel Hill
qhz991029@cs.unc.edu

**Tianlong Chen**
UNC Chapel Hill
tianlong@cs.unc.edu

## Abstract

With the advancing reasoning capabilities of Large Language Models (LLMs), they are increasingly employed for complex evaluation tasks, such as grading student responses, verifying factual claims, and comparing competing answers. Leveraging multiple LLMs as automated judges can enhance robustness and accuracy by aggregating diverse perspectives, yet existing approaches often rely on static and simple aggregation methods, such as majority voting, which may produce incorrect judgments despite correct individual assessments. We propose a novel multi-agent debate framework where LLMs collaboratively reason and iteratively refine judgments, formalizing this process mathematically and proving its advantages over static ensembles. To ensure computational efficiency, we introduce a stability detection mechanism using a time-varying Beta-Binomial mixture model (a mixture of two Beta-Binomial distributions) that tracks judge consensus dynamics and applies adaptive stopping via Kolmogorov–Smirnov testing. Experiments across diverse benchmarks and models demonstrate significant improvements in judgment accuracy over majority voting while maintaining computational efficiency.

## 1 Introduction

The rapid advancement of Large Language Models (LLMs) has significantly transformed automated evaluation, enabling near-human accuracy in assessing textual outputs [Chiang and Lee, 2023]. LLMs are now widely used for tasks such as scoring student essays for coherence [Xiao et al., 2025], fact-checking against reliable sources [Quelle and Bovet, 2024, Augenstein et al., 2024], and ranking multiple-choice answers for accuracy [Robinson and Wingate, 2023, Zheng et al., 2024], supporting applications in education [Wang et al., 2024b], content moderation, and decision support. A prominent approach in this context is the LLM-as-a-Judge paradigm [Zheng et al., 2023, Qu et al., 2025], where LLMs evaluate responses generated by other LLMs or humans. However, relying on a single LLM can be limiting due to potential biases and correlated errors [Tumer and Ghosh, 1996, Wang et al., 2023, 2025b]. To address these issues, multi-agent ensembles have been proposed [Li et al., 2024], which aggregate multiple LLM judgments through methods like weighted voting [Dietterich, 2000], averaging, stacking, and majority voting [Zhou, 2012].

Despite its simplicity, majority voting can be unreliable in complex or ambiguous cases, particularly when agents share similar biases or when the correct answer is a minority opinion [Yang et al., 2025]. This motivates the need for more robust frameworks that can capture the collective intelligence of multiple agents without being constrained by static aggregation methods. To address this, we propose a *multi-agent debate judge* framework as shown in Figure 1, where multiple LLMs engage

in structured debates to collaboratively reason and refine their judgments. We also present a formal mathematical model of the debate process, capturing agent interactions and belief updates. Building on this foundation, we prove that debate improves correctness over static ensembles under mild assumptions, establishing a theoretical basis for iterative refinement.

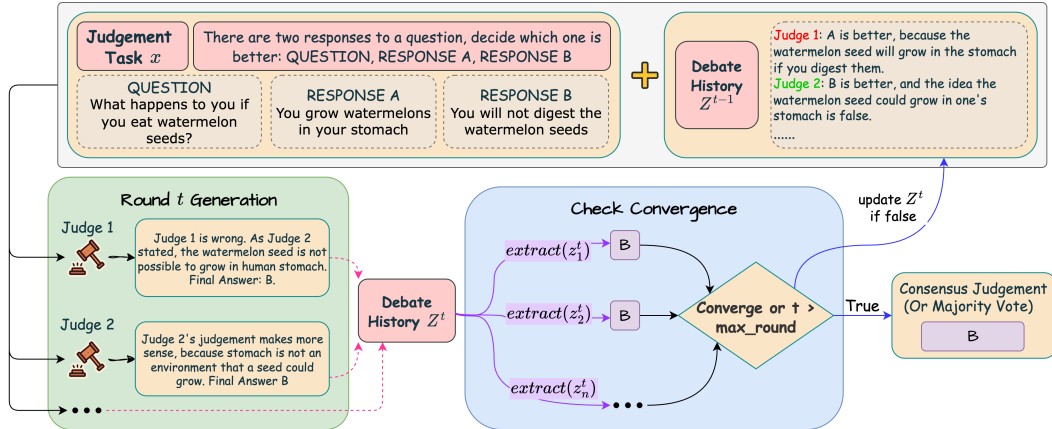

Figure 1: Multi-Agent Debate Framework.

However, iterative debates can be computationally expensive, especially when the process is not optimally terminated. Fixed-round debates risk either premature stopping before consensus is reached or unnecessary computation after convergence. To address this, we introduce a stability detection mechanism based on a time-varying mixture of Beta-Binomial distributions, using the Kolmogorov-Smirnov (KS) statistic [Massey Jr, 1951] to adaptively detect when the distribution stabilizes and to terminate the debate.

We validate our framework through experiments across diverse benchmarks, LLM architectures, and modalities (visual and non-visual tasks), demonstrating that our multi-agent debate framework outperforms majority voting in terms of accuracy, and the adaptive stopping mechanism significantly reduces computational costs while maintaining high accuracy.

**Our contributions:** (1) A formal debate framework for LLM ensembles that enables collaborative reasoning with theoretically provable correctness guarantees; (2) A novel stability detection mechanism using Beta-Binomial mixture modeling and adaptive stopping; (3) Comprehensive empirical validation showing substantial accuracy gains over majority voting.

## 2 Related Work

**LLMs-as-Judges.** Our work is closely related to the field of LLM-as-a-Judge [Zheng et al., 2023, Gu et al., 2025] and LLMs-as-Judges [Li et al., 2024], which involves using one or more LLMs to evaluate responses generated by either another LLM or a human. Basic LLM-as-Judge frameworks typically rely on a single LLM to perform a judgement task [Liu et al., 2023, Dubois et al., 2024]. Recent studies leverage LLMs to model user preferences or assess quality criteria [Shankar et al., 2024, Pan et al., 2024, Tian et al., 2024], judge factual consistency or hallucinations [Lin et al., 2022, Chen et al., 2024d, Luo et al., 2024], flag biased or unsafe content [Chen and Goldfarb-Tarrant, 2025, Yuan et al., 2024], and evaluate reasoning quality [Lightman et al., 2023, Srivastava et al., 2023]. However, LLM-based judges also exhibit several limitations [Koo et al., 2024, Wang et al., 2024a, Wu and Aji, 2025], such as self-preference bias [Wataoka et al., 2024], societal biases [Chen et al., 2024b], inconsistency [Stureborg et al., 2024], and other common challenges faced by LLMs [Dai, 2024].

**Multi-Agent Debate.** Recent work has explored multi-agent debate frameworks, where multiple agents engage in structured reasoning to reach consensus [Pham et al., 2024, Rasal, 2024, Michael et al., 2023, Chang, 2025, Irving et al., 2018, Khan et al., 2024, Du et al., 2024, Liang et al., 2024, Chan et al., 2024, Wang et al., 2025a, Lei et al., 2025]. Inspired by Minsky [1986], Du et al. [2024] proposed a framework in which multiple LLMs respond to a question independently, then refine

their answers after being shown responses from other agents. Estornell and Liu [2024] extended this concept by formalizing the debate process as an optimization problem, laying emphasis on the role of latent concepts—the underlying abstractions that drive both human and LLM-generated language [Xie et al., 2022, Jiang, 2023].

To enhance the debate process, researchers have incorporated methods such as chain-of-thought reasoning [Kojima et al., 2022, Wei et al., 2022], self-reflection [Ren et al., 2023, Tan et al., 2025b], and self-consistency [Wang et al., 2023]. Other studies have explored diverse debate strategies, including adversarial settings—where agents take opposing sides and a third agent acts as judge [Liang et al., 2024]— and collaborative approaches, where agents work together to iteratively solve a problem [Li et al., 2025a, Estornell et al., 2025].

**Statistical Approaches.**   To estimate the correctness of debate judges, Qu et al. [2025] proposed modeling judge correctness dynamics using a mixture of Beta-Binomial distributions, effectively capturing features such as bimodal peaks in the correctness distribution than traditional binomial models. The Expectation-Maximization (EM) algorithm [Moon, 1996] is commonly employed to estimate parameters in such mixture models [Sun et al., 2024, Qu et al., 2025]. For stability detection, an approach to monitor the distributional similarity of judge correctness is the Kolmogorov-Smirnov (KS) test [Massey Jr, 1951], which quantifies the maximum difference between two empirical cumulative distribution functions (CDFs).

# 3    Multi-Agent Debate Framework

In this section, we introduce the multi-agent debate framework for LLM judges. We begin by defining some important notations and the debate process: let $x$ be the task and $y$ the ground truth answer. Each of the $n$ agents is parameterized by $\phi_i \in \Phi$. Agent $i$'s response at round $t$ is $z_i^{(t)}$, with $e(z_i^{(t)})$ extracting its judgment. All responses at round $t$ form $Z^{(t)} = z_1^{(t)}, ..., z_n^{(t)}$. $T$ is the maximum rounds of debate.

## 3.1    Debate Process

The multi-agent debate framework involves $n$ agents, each parameterized by $\phi_i$: (1) At round 0, each agent receives task $x$ and generates an initial response $z_i^{(0)}$. (2) In each subsequent round, agents observe the task and debate history, then generate new responses. (3) After each round, if all agents agree, the process terminates and returns the consensus; otherwise, it continues until a maximum of $T$ rounds, after which the majority vote is returned. This procedure is summarized in Algorithm 1 in the appendix.

## 3.2    Latent Concepts

Following prior work [Xie et al., 2022, Jiang, 2023, Estornell and Liu, 2024], we adopt the notion of *latent concepts*, which refers to the underlying abstract ideas or interpretations that guide how agents understand and respond to a task.

Let $\Theta$ denote a latent concept space, where each concept $\theta \in \Theta$ represents a coherent interpretation of task $x$. The task-answer pair $(x, y)$ is generated by first sampling a concept $\theta$, then drawing $(x, y) \sim D(\theta)$, where $D$ maps concepts to task-answer pairs. Upon observing $x$, agents infer a distribution over $\Theta$ and generate responses accordingly. Multiple valid concepts may exist, and agents may focus on different aspects. Although $\Theta$ is abstract, we use sentence embeddings to represent and compare concepts in practice.

To provide a more detailed example of how latent concepts can be used in the debate process, consider the following question: "*Who won the 2021 Formula 1 Drivers' Championship?*" to which the correct answer would be "*Max Verstappen*". The latent concept behind this task involves knowledge of the 2021 Formula 1 season and the fact that Verstappen won the championship. Sentence embeddings are able to effectively capture this semantic concept and enable agents to align or disagree based on such latent understanding.

### 3.3 Response Generation Mechanism

At round $t$, agent $i$ generates a response $z_i^{(t)}$ based on the task $x$, the history of responses $Z^t$, and its parameters $\phi_i$, modeled as:

$$\mathbb{P}_{\text{model}}\big(z_i^{(t+1)} \mid x, Z^t, \phi_i\big).$$

Introducing a latent concept space $\Theta$, this becomes:

$$\mathbb{P}_{\text{model}}\big(z_i^{(t+1)} \mid x, Z^t, \phi_i\big) = \sum_{\theta \in \Theta} \mathbb{P}\big(z_i^{(t+1)} \mid \theta, x, Z^t, \phi_i\big) \mathbb{P}(\theta \mid x, Z^t, \phi_i). \tag{1}$$

The first term is the likelihood of generating $z_i^{(t+1)}$ under concept $\theta$; the second is the agent's updated belief in $\theta$ after observing $x$ and $Z^t$.

We now introduce a key assumption that simplifies the modeling process:

**Assumption 3.1** (Conditional Independence on Latent Concepts). *For a given latent concept $\theta$, the probability of generating response $z_i^{(t+1)}$ is conditionally independent of both $Z^{(t)}$ and $x$, given $\theta$ and $\phi_i$:*

$$\mathbb{P}\big(z_i^{(t+1)} \mid \theta, x, Z^t, \phi_i\big) = \mathbb{P}\big(z_i^{(t+1)} \mid \theta, \phi_i\big).$$

This assumption implies that the generation $z_i^{(t+1)}$ of model $i$ is solely determined by the latent concept $\theta$ of the input task and the agent's parameters $\phi_i$. Again with the example mentioned earlier, the sentence embeddings that capture the semantic meaning of "Max Verstappen won the 2021 Formula 1 Drivers' Championship" are produced solely based on the latent concept $\theta$ and the agent's parameters $\phi_i$.

**Lemma 3.1** (Response Generation Model). *Under Assumption 3.1, the generation of a response by model $i$ at time $t + 1$ can be expanded with Bayesian inference:*

$$\mathbb{P}\big(z_i^{(t+1)} \mid x, Z^t, \phi_i\big) \;\propto\; \sum_{\theta \in \Theta} \mathbb{P}\big(z_i^{(t+1)} \mid \theta, \phi_i\big) \mathbb{P}\big(x \mid \theta, \phi_i\big) \mathbb{P}\big(\theta \mid \phi_i\big) \prod_{j=1}^{n} \mathbb{P}\big(z_j^t \mid \theta, \phi_i\big). \tag{2}$$

This formulation clarifies how agents incorporate others' responses into their posterior beliefs about the latent concept, enabling collaborative refinement of judgments. Through Bayesian inference, each agent updates its belief in $\theta$ by weighing the likelihood of the task $x$ and all responses $Z^t$ against its prior $\mathbb{P}(\theta \mid \phi_i)$. This iterative process helps correct individual errors—such as those from biased training data—by shifting beliefs toward the correct concept, thus improving ensemble accuracy and mitigating correlated errors seen in static aggregation methods [Tumer and Ghosh, 1996]. Modeling response generation probabilistically over a latent concept space supports robust, collective deliberation.

## 4 Theoretical Analysis

### 4.1 Assumptions

Our analysis rests on four core assumptions that formalize how latent concepts govern the debate dynamics. We motivate each assumption with practical intuition and highlight its implications and limitations.

**Assumption 4.1** (True Concept Predictiveness). *For all agents $i$, concepts $\theta' \neq \theta^*$, and rounds $t$:*

$$\mathbb{P}(e(z_i^{t+1}) = y \mid \theta^*, \phi_i) > \mathbb{P}(e(z_i^{t+1}) = y \mid \theta', \phi_i).$$

This assumption asserts that the true concept $\theta^*$ leads to more accurate predictions than any other incorrect concept. It captures the intuitive idea that there exists a best way to frame the task (e.g., a correct scientific theory or legal principle), and that responses generated under this framing are more likely to be correct. While it simplifies the space of possible misinterpretations and might weaken if the tasks suffer from high ambiguity cases, it enables rigorous analysis of concept-driven reasoning dynamics.

**Assumption 4.2** (Task-Concept Alignment). *The probability of observing task $x$ is higher given the true concept than any incorrect concept:*

$$\mathbb{P}(x \mid \theta^*, \phi_i) > \mathbb{P}(x \mid \theta', \phi_i) \quad \forall \theta' \neq \theta^*.$$

This reflects that task generation is not uniform across concepts—some tasks are more naturally aligned with specific latent interpretations. For example, a medical diagnosis task is more likely to arise under a medical concept than under a legal one. This assumption allows posterior inference over $\theta$ using Bayes' rule to favor $\theta^*$ as debate unfolds.

**Assumption 4.3** (Positive Concept Prior Beliefs). *All concepts have positive prior probability:*

$$\mathbb{P}(\theta \mid \phi_i) > 0 \quad \forall \theta \in \Theta, \forall i.$$

This ensures that no concept is ruled out a priori, a standard regularity condition in Bayesian models. It prevents agents from permanently excluding the true concept and models diversity in agents' initial beliefs, where even implausible concepts retain some weight.

**Assumption 4.4** (Independent Agent Responses). *Agent responses are conditionally independent given the latent concept $\theta$:*

$$\mathbb{P}(z_1^t, z_2^t, \ldots, z_n^t \mid \theta, \phi) = \prod_{j=1}^{n} \mathbb{P}(z_j^t \mid \theta, \phi_j).$$

This assumption simplifies belief aggregation by treating agent responses as independent signals once the concept is fixed. Although this may be violated if agents copy or reference one another, or share strong biases, it is reasonable in decentralized debate settings where responses are generated in parallel.

## 4.2 Theorems

We begin by defining key concepts used in our analysis:

- *True Concept*: $\theta^*$, the unique concept such that $(x, y) \sim D(\theta^*)$, i.e., the concept that maximizes the likelihood of generating the correct answer.

- *Response Consistency*: $c(z_j^t, \theta) := \mathbb{P}(z_j^t \mid \theta, \phi_j)$, denoting the likelihood of response $z_j^t$ under concept $\theta$ and parameters $\phi_j$.

- *Strong Consistency*: A response $z_j^t$ is $\theta^*$-strong if $c(z_j^t, \theta^*) > c(z_j^t, \theta')$ for all $\theta' \neq \theta^*$. This captures the idea that a response is most likely generated under the true concept.

We now present two main theorems:

**Theorem 4.1** (Consistent Response Amplification). *Let $Z_A^t$ be a set of responses where at least one response is **strongly consistent** with the true concept $\theta^*$, and $Z_B^t$ be a set of responses where no response is strongly consistent with $\theta^*$. Then:*

$$\mathbb{E}_i\left[\mathbb{P}(a(z_i^{t+1}) = y \mid x, Z_A^t, \phi_i)\right] > \mathbb{E}_i\left[\mathbb{P}(a(z_i^{t+1}) = y \mid x, Z_B^t, \phi_i)\right], \tag{3}$$

*where $\mathbb{E}_i$ is the expectation over agents $i$. That is, the presence of at least one strongly consistent response in round $t$ **increases** the expected correctness in round $t + 1$.*

See Appendix A.1 for the full proof. This theorem formalizes a central benefit of debate: even a single correct reasoning path can guide other agents toward better beliefs and improved future performance. It supports the value of curriculum learning and few-shot prompting in multi-agent reasoning.

We next extract a useful consequence:

**Lemma 4.1** (Accuracy Increases with Posterior Belief). *Under the assumptions of Theorem 4.1, the probability of an agent producing a correct answer increases with their posterior belief in the true concept:*

$$\mathbb{P}(e(z_i^t) = y) \uparrow \mathbb{P}(\theta^* \mid Z^{t-1}).$$

See Appendix A.2 for the proof. This follows directly from Bayesian updating: stronger belief in $\theta^*$ improves expected predictive accuracy. It formalizes the link between belief refinement and task performance.

To prove that debate outperforms static aggregation (e.g., majority vote), we introduce one final condition: at least one response in the first round must be generated under the true concept, to enable belief updating.

**Assumption 4.5** (Initial Seed of Correct Reasoning). *There exists at least one initial response generated via the correct concept: latent concepts represented by reasoning path.* $\exists z_i^{(0)}$ *with* $c(z_i^{(0)}, \theta^*) > c(z_i^{(0)}, \theta')$. *This ensures the debate has a valid starting point for belief updates.*

We now state our second main theorem:

**Theorem 4.2** (Debate Improvement over Majority Vote). *Under the preceding assumptions, the final accuracy of the debated outcome $D(Z^T)$ exceeds that of initial majority vote $MV(Z^0)$:*

$$\mathbb{P}(D(Z^T) = y) > \mathbb{P}(MV(Z^0) = y). \tag{4}$$

See Appendix A.3 for the full proof. This result supports the view that structured interaction—through iterative debate—enables a population of agents to converge on more accurate answers than independent majority voting. It aligns with classical findings in distributed reasoning and ensemble methods, where collaborative refinement outperforms static aggregation.

# 5 Debate Adaptive Stability Detection

To improve debate efficiency, we introduce an **adaptive stability detection mechanism** that halts the process once judge accuracy rates stabilize. We model judge accuracy as a time-varying Beta-Binomial mixture, estimating parameters via Expectation-Maximization (EM). Stability is detected by monitoring distributional similarity across rounds using the Kolmogorov–Smirnov (KS) statistic. See Algorithm 2 in the appendix.

## 5.1 Judgement Accuracy Modeling

Let $\psi_i$ denote the latent correct rate of a debate judge at round $i$, with distribution $D_i$. Our goal is to determine when $D_i$ stabilizes sufficiently to compute reliable bounds for $\psi_i$.

We observe an ensemble of $k$ judges whose collective decisions produce a score $S^t$ at each round $t$—the total number of correct decisions. We model $S^t$ as a time-varying mixture of two Beta-Binomial distributions:

$$S^t \sim w^t \, \text{BB}(k, \alpha_1^t, \beta_1^t) + (1 - w^t) \, \text{BB}(k, \alpha_2^t, \beta_2^t). \tag{5}$$

Here, $\text{BB}(k, \alpha, \beta)$ denotes the Beta-Binomial distribution, which models the number of correct decisions among $k$ judges with shape parameters $\alpha$ and $\beta$, capturing the variability in judge accuracy due to heterogeneous behaviors. The mixture weight $w^t \in [0, 1]$ balances the two components, and $\alpha_1^t, \beta_1^t, \alpha_2^t, \beta_2^t$ parameterize the two components. This model captures different behavioral regimes among judges (e.g., attentive vs. inattentive).

## 5.2 Parameter Estimation via Expectation-Maximization

For each round $t$, we estimate parameters $\psi^t = \{w^t, \alpha_1^t, \beta_1^t, \alpha_2^t, \beta_2^t\}$ from $n$ observed values $\{s_1^t, ..., s_n^t\}$ using maximum likelihood estimation with the EM algorithm. The complete-data likelihood combines both mixture components:

$$\mathcal{L}(\psi^t) = \prod_{j=1}^{n} \left[ w^t \text{BB}(s_j^t; k, \alpha_1^t, \beta_1^t) + (1 - w^t) \text{BB}(s_j^t; k, \alpha_2^t, \beta_2^t) \right], \tag{6}$$

where the Beta-Binomial probability mass function is defined as:

$$\text{BB}(s; k, \alpha, \beta) = \binom{k}{s} \frac{B(s + \alpha, k - s + \beta)}{B(\alpha, \beta)},$$

and $B(\alpha, \beta) = \frac{\Gamma(\alpha)\Gamma(\beta)}{\Gamma(\alpha+\beta)}$ is the Beta function, with $\Gamma$ denoting the Gamma function.

The EM algorithm iteratively refines estimates of $\psi^t$:

- **E-step:** Compute responsibilities $r_{j,1}^t = \frac{w^t \mathrm{BB}(s_j^t; \alpha_1^t, \beta_1^t)}{w^t \mathrm{BB}(s_j^t; \alpha_1^t, \beta_1^t) + (1-w^t)\mathrm{BB}(s_j^t; \alpha_2^t, \beta_2^t)}$.

- **M-step:** Update parameters using weighted MLEs (Maximum Likelihood Estimation):

$$w^t \leftarrow \frac{1}{n}\sum_{j=1}^n r_{j,1}^t \quad \text{and} \quad \{\alpha_c^t, \beta_c^t\} \leftarrow \arg\max_{\alpha,\beta} \sum_{j=1}^n r_{j,c}^t \log \mathrm{BB}(s_j^t; \alpha, \beta) \quad (c = 1, 2).$$

In practice, we employ the L-BFGS-B optimization method [Zhu et al., 1997] to update the Beta-Binomial parameters. The algorithm terminates when the log-likelihood improvement is less than a convergence threshold $\epsilon = 10^{-6}$, or after a maximum of $n = 100$ iterations. This threshold was chosen to ensure high precision in parameter estimation while maintaining computational efficiency, as validated in our experiments across benchmarks.

### 5.3 Stability Detection

After the EM algorithm converges, meaning the log-likelihood improvement falls below a threshold $\epsilon$ or a maximum of $n$ iterations is reached, it yields an estimated parameter set $\psi^t = \{w^t, \alpha_1^t, \beta_1^t, \alpha_2^t, \beta_2^t\}$ for round $t$. The distribution over individual judges' correct rates is then given by:

$$P^t(\psi) = w^t \mathrm{Beta}(\psi; \alpha_1^t, \beta_1^t) + (1-w^t)\mathrm{Beta}(\psi; \alpha_2^t, \beta_2^t), \tag{7}$$

where $\mathrm{Beta}(\psi; \alpha, \beta) = \frac{\psi^{\alpha-1}(1-\psi)^{\beta-1}}{B(\alpha,\beta)}$ is the probability density function of the Beta distribution, and $B(\alpha, \beta) = \frac{\Gamma(\alpha)\Gamma(\beta)}{\Gamma(\alpha+\beta)}$ is the Beta function defined in the previous subsection.

To detect when this distribution stabilizes, we track the Kolmogorov-Smirnov (KS) statistic between consecutive rounds:

$$D_t = \sup_{\psi \in [0,1]} |F^t(\psi) - F^{t-1}(\psi)|, \tag{8}$$

where $F^t$ is the cumulative distribution function (CDF) of $P^t(\psi)$. As described in Algorithm 2, the **judgement accuracy modeling** process halts once $D_t < 0.05$ for 2 consecutive rounds, as used in our experiments, signaling that the judge accuracy distribution has stabilized.

## 6 Experiments

### 6.1 Experimental Setup

Our evaluation framework assesses a wide range of state-of-the-art LLMs, including both proprietary and open-source models from multiple providers across visual and non-visual tasks. For the proprietary model, we use Gemini-2.0-Flash [Google, 2024] from Google. Open-source models comprise Llama-3.1-8B-Instruct [Grattafiori et al., 2024, Meta, 2024] and Llama-3.2-11B-Vision-Instruct [Grattafiori et al., 2024, AI, 2024], both from Meta AI, used for non-visual and visual tasks, respectively; Qwen-2.5-7B-Instruct [Qwen et al., 2025] and Qwen-2.5-VL-7B-Instruct [Bai et al., 2025], both from Alibaba, applied to non-visual and visual tasks, respectively; and Gemma-3-4B-Instruct [Team et al., 2025] from Google used for both tasks.

We conduct experiments on datasets from diverse domains to evaluate the debate judge's performance, including: *hallucination detection*: TruthfulQA [Lin et al., 2022], *alignment evaluation*: JudgeBench [Tan et al., 2025a] and LLMBar [Zeng et al., 2024], and *reasoning*: BIG-Bench [Srivastava et al., 2023]. We also use multiple multi-modal datasets: MLLM-Judge [Chen et al., 2024a] and JudgeAnything [Pu et al., 2025].

### 6.2 Comparative Results

Table 1 shows that our debate framework generally outperforms both baselines: Single Model and SoM (Majority Vote), especially on complex tasks like JudgeBench, LLMBar, TruthfulQA, and

|  | **BIG-Bench** | | | **JudgeBench** | | |
| Model | **Single** | **SoM** | **Debate** | **Single** | **SoM** | **Debate** |
|---|---|---|---|---|---|---|
| Gemma-3-4B | $69.84_{\pm2.45}$ | $70.80_{\pm2.81}$ | $\mathbf{71.10}_{\pm2.81}$ | $55.62_{\pm3.24}$ | $54.60_{\pm3.91}$ | $\mathbf{56.70}_{\pm3.89}$ |
| Qwen-2.5-7B | $74.37_{\pm2.10}$ | $\mathbf{76.60}_{\pm2.62}$ | $72.20_{\pm2.77}$ | $58.32_{\pm2.93}$ | $59.52_{\pm3.85}$ | $\mathbf{59.68}_{\pm3.85}$ |
| Llama-3.1-8B | $78.67_{\pm1.94}$ | $\mathbf{81.80}_{\pm2.39}$ | $74.00_{\pm2.72}$ | $57.98_{\pm3.02}$ | $\mathbf{60.84}_{\pm3.84}$ | $58.90_{\pm3.87}$ |
| Gemini-2.0-Flash | $81.74_{\pm2.16}$ | $81.50_{\pm2.41}$ | $\mathbf{82.30}_{\pm2.36}$ | $63.66_{\pm3.03}$ | $66.13_{\pm3.72}$ | $\mathbf{68.06}_{\pm3.66}$ |

|  | **LLMBar** | | | **TruthfulQA** | | |
| Model | **Single** | **SoM** | **Debate** | **Single** | **SoM** | **Debate** |
|---|---|---|---|---|---|---|
| Gemma-3-4B | $57.98_{\pm2.48}$ | $57.83_{\pm2.79}$ | $\mathbf{58.83}_{\pm2.78}$ | $40.39_{\pm2.99}$ | $40.15_{\pm3.38}$ | $\mathbf{41.62}_{\pm3.37}$ |
| Qwen-2.5-7B | $65.57_{\pm2.21}$ | $66.22_{\pm2.67}$ | $\mathbf{69.81}_{\pm2.60}$ | $59.84_{\pm2.86}$ | $\mathbf{62.39}_{\pm3.36}$ | $58.51_{\pm3.37}$ |
| Llama-3.1-8B | $59.70_{\pm2.36}$ | $60.25_{\pm2.76}$ | $\mathbf{62.58}_{\pm2.73}$ | $50.83_{\pm2.85}$ | $53.94_{\pm3.48}$ | $\mathbf{55.34}_{\pm3.41}$ |
| Gemini-2.0-Flash | $76.68_{\pm1.97}$ | $77.75_{\pm2.35}$ | $\mathbf{81.83}_{\pm2.18}$ | $69.49_{\pm2.71}$ | $72.01_{\pm3.10}$ | $\mathbf{74.30}_{\pm2.99}$ |

|  | **MLLM-Judge** | | | **JudgeAnything** | | |
| Model | **Single** | **SoM** | **Debate** | **Single** | **SoM** | **Debate** |
|---|---|---|---|---|---|---|
| Gemma-3-4B | $61.13_{\pm3.04}$ | $61.62_{\pm3.36}$ | $\mathbf{62.75}_{\pm3.34}$ | $83.46_{\pm5.81}$ | $\mathbf{84.96}_{\pm6.07}$ | $\mathbf{84.96}_{\pm6.07}$ |
| Qwen-2.5-VL-7B | $60.43_{\pm3.27}$ | $\mathbf{60.88}_{\pm3.37}$ | $60.38_{\pm3.38}$ | $67.88_{\pm7.84}$ | $\mathbf{68.42}_{\pm3.37}$ | $67.67_{\pm7.85}$ |
| Gemini-2.0-Flash | $67.50_{\pm2.88}$ | $68.00_{\pm3.23}$ | $\mathbf{69.25}_{\pm3.19}$ | $81.63_{\pm5.70}$ | $83.46_{\pm6.30}$ | $\mathbf{85.71}_{\pm5.95}$ |

Table 1: Accuracy (%) and standard error (%) of different response aggregation methods—Single (sampling once), SoM (Majority Vote), and Debate (10 Rounds Maximum)—across datasets and models. All results use an ensemble size of 7 and a sampling temperature of 1.0.

MLLM-Judge. Gemini-2.0-Flash achieves the largest gains in several cases (e.g., 77.75% to 81.83% on LLMBar). These gains are modest in some cases because our framework's iterative refinement adds most value in complex tasks with high initial variance, where collaborative belief updates correct biases (Theorem 4.1), yielding significant improvements. On simpler tasks with high initial consensus, such as BIG-Bench and JudgeAnything, SoM performs comparably or better as refinement introduces minimal benefit, aligning with diminishing returns in low-variance scenarios. This supports targeted applicability: debate excels where accuracy justifies costs, while SoM suffices for straightforward tasks.

Our analysis (Table 5, Appendix B.2) shows that an ensemble size of 7 provides the best balance between accuracy and computational cost across most tasks. Larger ensembles (Size-9 or greater) show diminishing returns in accuracy, while increasing computational costs, smaller ensembles (Size-5) are sufficient to maintain accuracy with minimal cost. We recommend Size-7 as the optimal choice for most use cases.

### 6.3 Judgement Dynamics

**Judgement Distribution.** Figure 2a shows the evolution of correct agent distributions across debate rounds on JudgeBench for four models. Initially, Round 0 distributions are broad, reflecting diverse judgments. By Rounds 2, distributions converge to a bimodal pattern (0 or 7 correct agents), maintaining a Beta-Binomial mixture shape, indicating that agents either align on the correct answer or collectively fail. Similar convergence is observed across other datasets (see Appendix B.2.1), confirming the debate framework's robustness.

Figure 2b illustrates the distribution of correct agents across debate rounds for the Llama-3.1-8B model on the JudgeBench dataset. The solid line represents the fitted Beta-Binomial distribution, while shaded areas depict the empirical distribution of correct agents (x-axis) with probability density (y-axis). The close alignment between the fitted and empirical distributions highlights the effectiveness of the EM algorithm in modeling agent performance dynamics.

**Adaptive Stability Detection.** Figure 3 presents KS statistics across six debate rounds for six datasets and five models. The KS statistic (y-axis) measures the difference between CDFs of correct agent counts across two consecutive rounds (x-axis). High initial KS values (e.g., 0.25–0.45 for JudgeBench, Round 1) reflect diverse judgments and opinion changing, but values typically rapidly drop below the stability threshold ($\epsilon = 0.05$) within 2 to 7 rounds (e.g., Gemini-2.0-Flash on BIG-Bench by Round 2). To prevent premature halting, the adaptive mechanism requires KS values

|  | BIG-Bench | | | JudgeBench | | |
|---|---|---|---|---|---|---|
| Model | Rounds | Accuracy | Diff | Rounds | Accuracy | Diff |
| Gemma-3-4B | 5 | $70.07_{\pm 2.82}$ | -1.03 | 5 | $56.54_{\pm 3.89}$ | -0.16 |
| Qwen-2.5-7B | 7 | $72.00_{\pm 2.78}$ | -0.20 | 6 | $59.35_{\pm 3.85}$ | -0.33 |
| Llama-3.1-8B | 7 | $73.70_{\pm 2.73}$ | -0.30 | 6 | $58.58_{\pm 3.87}$ | -0.32 |
| Gemini-2.0-Flash | 4 | $81.70_{\pm 2.40}$ | -0.60 | 6 | $67.74_{\pm 3.70}$ | -0.32 |

|  | LLMBar | | | TruthfulQA | | |
|---|---|---|---|---|---|---|
| Model | Rounds | Accuracy | Diff | Rounds | Accuracy | Diff |
| Gemma-3-4B | 5 | $58.75_{\pm 2.78}$ | -0.08 | 5 | $41.49_{\pm 3.37}$ | -0.13 |
| Qwen-2.5-7B | 5 | $69.14_{\pm 2.61}$ | -0.67 | 5 | $58.02_{\pm 3.37}$ | -0.49 |
| Llama-3.1-8B | 6 | $62.17_{\pm 2.74}$ | -0.41 | 6 | $54.72_{\pm 3.41}$ | -0.62 |
| Gemini-2.0-Flash | 5 | $81.33_{\pm 2.20}$ | -0.50 | 5 | $73.81_{\pm 3.01}$ | -0.49 |

|  | MLLM-Judge | | | JudgeAnything | | |
|---|---|---|---|---|---|---|
| Model | Rounds | Accuracy | Diff | Rounds | Accuracy | Diff |
| Gemma-3-4B | 4 | $62.50_{\pm 3.35}$ | -0.25 | 2 | $84.96_{\pm 6.07}$ | 0.00 |
| Qwen-2.5-VL-7B | 4 | $60.38_{\pm 3.38}$ | 0.00 | 2 | $67.67_{\pm 7.85}$ | 0.00 |
| Gemini-2.0-Flash | 5 | $68.63_{\pm 3.21}$ | -0.62 | 8 | $85.71_{\pm 5.95}$ | 0.00 |

Table 2: Adaptive stopping performance in the Debate method: number of rounds until stopped, accuracy (%), and accuracy difference (%) compared to using the full 10 rounds. All experiments use an ensemble size of 7, a maximum of 10 debate rounds and a KS-statistic threshold of 0.05.

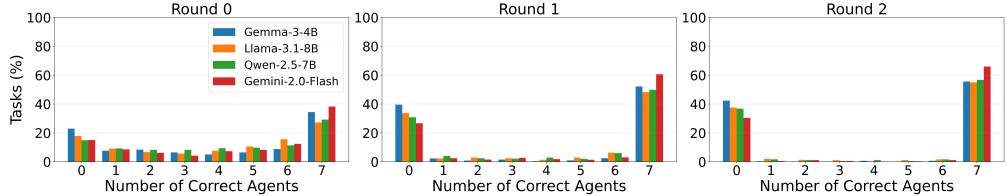

(a) Distribution of correct agents across 3 debate rounds on JudgeBench for multiple models. Each subplot shows a round, with distributions converging to either 0 or 7 correct agents, reflecting the debate process's alignment effect.

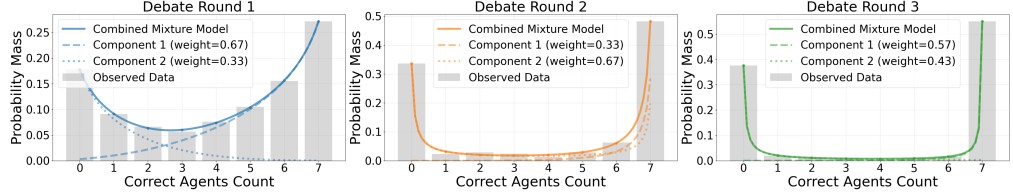

(b) Fitted Beta-Binomial distributions (solid lines) against empirical distributions (shaded areas) for Llama-3.1-8B on JudgeBench across debate rounds, showing the accuracy of our mixture model.

Figure 2: Judge consensus dynamics during debate. Top: Correct agent distributions across three rounds on JudgeBench, showing convergence to unanimous agreement. Bottom: Fitted Beta-Binomial mixture model closely matches empirical distributions for Llama-3.1-8B.

to remain below $\epsilon = 0.05$ for two consecutive rounds before terminating the debate process. For example, Gemini-2.0-Flash on JudgeAnything drops below this threshold by Round 3 but bounces back, until finally stabilizing from Round 6 onward.

## 7 Conclusion

In this paper, we introduced a multi-agent debate framework that allows LLMs to collaboratively reason and iteratively refine their judgments, addressing the shortcomings of static aggregation methods such as majority voting. Central to our approach is a novel stability detection mechanism, which employs a time-varying Beta-Binomial mixture model and the Kolmogorov–Smirnov statistic to

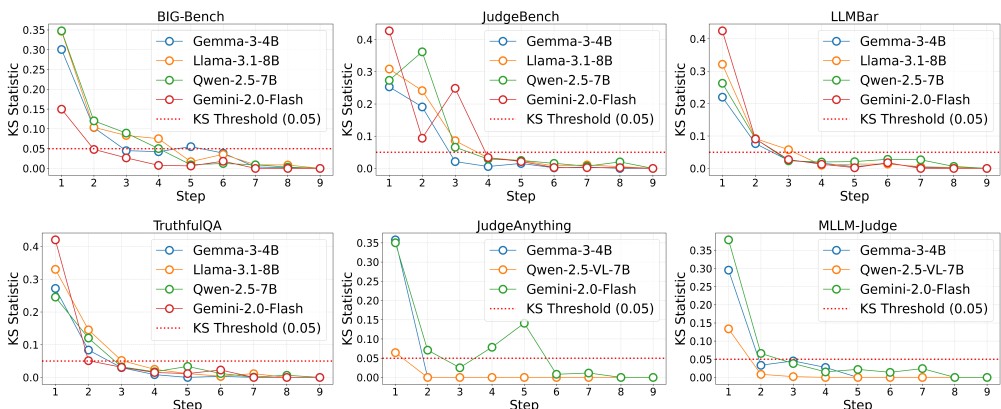

Figure 3: KS statistics across ten debate rounds for six datasets. The x-axis shows steps between rounds, the y-axis shows KS values, and the red dotted line marks the stability threshold ($\epsilon = 0.05$).

adaptively halt the debate process when consensus is achieved. The significance of our framework lies in its ability to bolster the robustness and precision of LLM-based evaluations through collaborative reasoning and iterative refinement. The stability detection mechanism optimizes resource use, making it viable for practical applications.

## Acknowledgments

This research is supported in part by an Amazon Research Award and a Cisco Faculty Award.

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

# A  Theoretical Analysis

## A.1  Proof of Theorem 4.1

We prove that having at least one strongly consistent response in round $t$ increases the expected probability of correctness in round $t + 1$. This relies on assumptions 4.1–4.4.

*Proof.* Using Bayes' rule and our defined assumptions, we can express the probability that agent $i$ generates the correct answer in round $t + 1$ as:

$$\mathbb{P}(e(z_i^{t+1}) = y \mid x, Z^t, \phi_i) = \sum_{\theta \in \Theta} \mathbb{P}(e(z_i^{t+1}) = y \mid \theta, \phi_i)\mathbb{P}(\theta \mid x, Z^t, \phi_i) \tag{9}$$

The posterior probability of concept $\theta$ given the observed responses $Z^t$ can be calculated as:

$$\mathbb{P}(\theta \mid x, Z^t, \phi_i) = \frac{\mathbb{P}(Z^t \mid \theta, x, \phi_i)\mathbb{P}(\theta \mid x, \phi_i)}{\mathbb{P}(Z^t \mid x, \phi_i)} \tag{10}$$

$$= \frac{\mathbb{P}(Z^t \mid \theta, \phi_i)\mathbb{P}(x \mid \theta, \phi_i)\mathbb{P}(\theta \mid \phi_i)}{\mathbb{P}(Z^t \mid x, \phi_i)\mathbb{P}(x \mid \phi_i)} \tag{11}$$

$$\propto \mathbb{P}(Z^t \mid \theta, \phi_i)\mathbb{P}(x \mid \theta, \phi_i)\mathbb{P}(\theta \mid \phi_i) \tag{12}$$

By assumption 4.4, we have:

$$\mathbb{P}(Z^t \mid \theta, \phi_i) = \prod_{j=1}^{n} \mathbb{P}(z_j^t \mid \theta, \phi_j) \tag{13}$$

This gives us:

$$\mathbb{P}(\theta \mid x, Z^t, \phi_i) \propto \mathbb{P}(x \mid \theta, \phi_i)\mathbb{P}(\theta \mid \phi_i) \prod_{j=1}^{n} \mathbb{P}(z_j^t \mid \theta, \phi_j) \tag{14}$$

Now, consider two sets of responses $Z_A^t$ and $Z_B^t$, where $Z_A^t$ contains at least one strongly consistent response with $\theta^*$ and $Z_B^t$ contains none. Let $z_s^t \in Z_A^t$ be this strongly consistent response.

By definition of strong consistency, $\mathbb{P}(z_s^t \mid \theta^*, \phi_s) > \mathbb{P}(z_s^t \mid \theta', \phi_s)$ for all $\theta' \neq \theta^*$.

For the sets $Z_A^t$ and $Z_B^t$, we have:

$$\frac{\mathbb{P}(\theta^* \mid x, Z_A^t, \phi_i)}{\mathbb{P}(\theta' \mid x, Z_A^t, \phi_i)} = \frac{\mathbb{P}(x \mid \theta^*, \phi_i)\mathbb{P}(\theta^* \mid \phi_i)}{\mathbb{P}(x \mid \theta', \phi_i)\mathbb{P}(\theta' \mid \phi_i)} \cdot \frac{\prod_{j=1}^{n} \mathbb{P}(z_j^t \mid \theta^*, \phi_j)}{\prod_{j=1}^{n} \mathbb{P}(z_j^t \mid \theta', \phi_j)} \tag{15}$$

$$> \frac{\mathbb{P}(x \mid \theta^*, \phi_i)\mathbb{P}(\theta^* \mid \phi_i)}{\mathbb{P}(x \mid \theta', \phi_i)\mathbb{P}(\theta' \mid \phi_i)} \cdot \frac{\prod_{j \neq s} \mathbb{P}(z_j^t \mid \theta^*, \phi_j)}{\prod_{j \neq s} \mathbb{P}(z_j^t \mid \theta', \phi_j)} \tag{16}$$

By assumption 4.2, we have $\mathbb{P}(x \mid \theta^*, \phi_i) > \mathbb{P}(x \mid \theta', \phi_i)$. Combined with the above, this shows that:

$$\mathbb{P}(\theta^* \mid x, Z_A^t, \phi_i) > \mathbb{P}(\theta^* \mid x, Z_B^t, \phi_i) \tag{17}$$

Using assumption 4.1, we can then show:

$$\mathbb{E}_i\big[\mathbb{P}(e(z_i^{t+1}) = y \mid x, Z_A^t, \phi_i)\big] > \mathbb{E}_i\big[\mathbb{P}(e(z_i^{t+1}) = y \mid x, Z_B^t, \phi_i)\big] \tag{18}$$

This completes the proof of Theorem 4.1. $\qquad\square$

## A.2  Proof of Lemma 4.1

We are given that each agent chooses their judgment by maximizing expected correctness based on their belief distribution over concepts:

$$\mathbb{P}(e(z_i^t) = y) = \sum_{\theta} \mathbb{P}(e(z_i^t) = y \mid \theta, \phi_i) \cdot \mathbb{P}(\theta \mid Z^{t-1}). \tag{19}$$

Since $\theta^*$ is the true concept (i.e., it produces the correct label $y$ with the highest probability), and agent reasoning reliability is fixed (via $\phi_i$), we assume:

$$\mathbb{P}(e(z_i^t) = y \mid \theta^*, \phi_i) > \mathbb{P}(e(z_i^t) = y \mid \theta\prime, \phi_i), \quad \forall \theta\prime \neq \theta^*. \tag{20}$$

Then, as $\mathbb{P}(\theta^* \mid Z^{t-1})$ increases (due to observing consistent responses), the overall weighted sum increases:

$$\Rightarrow \mathbb{P}(e(z_i^t) = y) \uparrow \mathbb{P}(\theta^* \mid Z^{t-1}), \tag{21}$$

establishing the claim.

## A.3  Proof of Theorem 4.2

We now show that the entire iterative debate process yields better outcomes than a simple majority vote on the initial responses. This result relies on Theorem A.1, the assumption 4.5, lemma 4.1, and lemma 3.1.

*Proof.* We first define the accuracy at round $0$ as

$$Acc(0) = \frac{1}{n} \sum_{i=1}^{n} \mathbb{P}(e(z_i^0) = y \mid x, \phi_i). \tag{22}$$

By standard concentration bounds (or accounting for ties/correlations), the probability that the initial majority vote matches the correct answer $y$ can be bounded as

$$\mathbb{P}(MV(Z^0) = y) \leq Acc(0) + \epsilon_0, \tag{23}$$

where $\epsilon_0 \geq 0$ captures minor discrepancies.

Next, at each round $t \geq 0$, each agent $i$ updates its posterior $\mathbb{P}(\theta \mid x, Z^t, \phi_i)$ using Bayes' rule 3.1. Under lemma 4.1, if $\mathbb{P}(\theta^* \mid x, Z^t, \phi_i)$ increases, then the agent's probability of producing the correct answer at round $t+1$ also increases. From Theorem A.1, any round $t$ containing at least one strongly consistent response with $\theta^*$ pushes beliefs further toward $\theta^*$. Because assumption 4.5 guarantees a strongly consistent response already at $t = 0$, it follows inductively that

$$Acc(t+1) = \frac{1}{n} \sum_{i=1}^{n} \mathbb{P}(e(z_i^{t+1}) = y \mid x, Z^t, \phi_i) > Acc(t), \quad \text{for all } t \geq 0. \tag{24}$$

Thus, repeated updates strictly increase the ensemble accuracy from one round to the next.

Iterating inequality (24) from $t = 0$ up to $t = T-1$ gives

$$Acc(T) > Acc(0). \tag{25}$$

Finally, the debate outcome $D(Z^T)$ is the majority vote at round $T$. Let $\epsilon_T \geq 0$ denote residual discrepancies from ties/correlation among agents at the final round. We then have

$$\mathbb{P}(D(Z^T) = y) \geq Acc(T) - \epsilon_T. \tag{26}$$

Combining (25) and (26), and comparing with the initial majority-vote probability in (23), we conclude:

$$\mathbb{P}(D(Z^T) = y) > \mathbb{P}(MV(Z^0) = y) - (\epsilon_0 + \epsilon_T).$$

In practice, $\epsilon_0$ and $\epsilon_T$ become negligible for large $n$ or well-calibrated agents, implying

$$\mathbb{P}(D(Z^T) = y) > \mathbb{P}(MV(Z^0) = y).$$

Hence, an iterative multi-agent debate outperforms a single-round majority vote, completing the proof of Theorem 4.2. □

# B Experimental Details

## B.1 Additional Dataset Details

We evaluate our framework on a diverse set of benchmarks spanning language understanding, instruction following, truthfulness, and multi-modal judgment:

- **BIG-Bench** [Srivastava et al., 2023]: A large-scale suite designed to test LLM capabilities across a wide range of tasks and domains. For efficiency and relevance, we focus on a curated subset of *sports understanding* tasks, each requiring models to determine the plausibility of given statements.

- **LLMBar** [Zeng et al., 2024]: A benchmark for instruction-following, containing 419 instances. Each instance presents an instruction, two candidate responses, and a label indicating which response is better. We use all available instances.

- **TruthfulQA** [Lin et al., 2022]: Designed to assess the truthfulness of LLMs, this benchmark includes over 800 questions, each with multiple correct and incorrect answers. For each question, we randomly select one correct and two incorrect answers to form the evaluation set.

- **JudgeBench** [Tan et al., 2025a]: Focused on judgment and alignment, this dataset provides 620 response pairs, each labeled to indicate which response is better.

- **MLLM-Judge** [Chen et al., 2024a]: A multi-modal benchmark evaluating judgment in visual tasks. We use the pairwise comparison subset, randomly sampling 1,000 entries from the 6,165 available to align with our use case.

- **JudgeAnything** [Pu et al., 2025]: A multi-modal benchmark covering text, image, audio, and video. We evaluate on the image-to-text pairwise comparison subset, which contains 180 entries.

This selection ensures comprehensive coverage of both textual and multi-modal evaluation scenarios, enabling robust assessment of our debate framework across diverse tasks and modalities.

## B.2 Additional Experiments Details

**Hyperparameters.** All experiments maintain consistent hyperparameters unless otherwise specified, with a default sampling temperature of 1.0 to balance response diversity and coherence. Ensemble size is set to 7, and the maximum debate rounds are capped at 10. The max model length for all models was set to 16,000 tokens.

**Multi-Agent Debate Process** The multi-agent debate process is outlined in Algorithm 1.

**Adaptive Stopping Mechanism** The adaptive stopping mechanism is outlined in Algorithm 2.

We evaluated Gemini-2.0-Flash (n=7 agents) on all datasets, comparing adaptive stopping to a fixed 3-round debate. Results, shown in Table 3, demonstrate that adaptive stopping achieves comparable or better accuracy while using fewer rounds on average. Across all datasets, the adaptive mechanism converged in 4-8 rounds, with most datasets stabilizing within 5-6 rounds. The accuracy improvements, while modest (ranging from 0.1% to 0.6%), come with the benefit of computational efficiency—the adaptive approach processes only the necessary rounds rather than a fixed number.

To analyze the sensitivity of the stopping criterion, we conducted an ablation study varying the KS threshold $\epsilon$ on the JudgeBench dataset. Table 4 shows how different threshold values affect stopping behavior. Lower thresholds (e.g., 0.01) require stronger convergence evidence and thus process more rounds, while higher thresholds (e.g., 0.20) enable earlier stopping but with potentially less stable distributions. The results indicate a practical sweet spot between 0.05 and 0.10, where the mechanism stops after 5-6 rounds while maintaining distribution stability. This demonstrates that the adaptive stopping parameters can be tuned to balance accuracy and computational cost based on specific application requirements.

**Affect of Ensemble Size on Debate.** Table 5 and Table 6 collectively illustrate the trade-off between accuracy and computational cost in the Debate method for the Gemma-3-4B model across different ensemble sizes and benchmarks. Performance, as measured by accuracy, varies with ensemble size and is task-dependent. For most benchmarks, including BIG-Bench, JudgeBench,

**Algorithm 1** Multi-Agent Debate Process

---

**Require:** Input task $x$, agents $\{\phi_i\}_{i=1}^n$, max rounds $T$
**Ensure:** Ground truth $y$
 1: Initialize $Z^{(0)} \leftarrow \emptyset$
 2: **for** each agent $i \in 1, \ldots, n$ **do**
 3:     Generate initial response: $z_i^{(0)} \sim P_{\text{init}}(x|\phi_i)$
 4:     Update history: $Z^{(0)} \leftarrow Z^{(0)} \cup \{z_i^{(0)}\}$
 5: **end for**
 6: **for** $t = 1$ **to** $T$ **do**
 7:     **for** each agent $i \in 1, \ldots, n$ **in parallel do**
 8:         Observe history: $Z^{(t-1)}$
 9:         Generate response: $z_i^{(t)} \sim P_{\text{resp}}(x, Z^{(t-1)}|\phi_i)$
10:         Update history: $Z^{(t)} \leftarrow Z^{(t)} \cup \{z_i^{(t)}\}$
11:     **end for**
12:     Compute consensus: $c^{(t)} \leftarrow \text{mode}(\{e(z_1^{(t)}), ..., e(z_n^{(t)})\})$
13:     **if** unanimous($c^{(t)}$) **then**
14:         **return** $c^{(t)}$                                    ▷ Early termination on consensus
15:     **end if**
16: **end for**
17: **return** majority($\{e(z_1^{(T)}), ..., e(z_n^{(T)})\}$)

---

**Algorithm 2** Adaptive Stopping Mechanism

---

**Require:** Judges $k$, threshold $\epsilon$, stability rounds $m$
 1: Initialize $t \leftarrow 1$, $c \leftarrow 0$
 2: Collect samples $\{s_1^1, ..., s_n^1\}$
 3: Estimate $\psi^1$ via EM; compute $P^1(\psi)$
 4: **while** not stopped **do**
 5:     $t \leftarrow t + 1$
 6:     Collect $\{s_1^t, ..., s_n^t\}$; estimate $\psi^t$
 7:     Compute $P^t(\psi) = w^t \text{Beta}(\alpha_1^t, \beta_1^t) + (1 - w^t)\text{Beta}(\alpha_2^t, \beta_2^t)$
 8:     Evaluate KS statistic $D_t$ on $[0, 1]$ grid
 9:     **if** $D_t < \epsilon$ **then**
10:         $c \leftarrow c + 1$
11:         **if** $c \geq m$ **then**
12:             **Stop**
13:         **end if**
14:     **else**
15:         $c \leftarrow 0$
16:     **end if**
17:     Update $P^{t-1} \leftarrow P^t$
18: **end while**
19: **return** Final round $t$, parameters $\psi^t$

---

LLMBar, MLLM-Judge, and JudgeAnything, the highest accuracy is achieved with an ensemble size of 7 (e.g., 71.10±2.81 for BIG-Bench, 84.96±6.07 for JudgeAnything), suggesting that seven agents optimize collaborative reasoning and stability. In contrast, TruthfulQA peaks at Size-5 (41.86±3.38) and remains stable across larger sizes, likely due to its simpler nature, as seen in questions like "What happens to you if you eat watermelon seeds?" which require minimal reasoning. Computational cost, reflected in token counts from Table 6, rises with ensemble size, with demanding tasks like JudgeBench (1321.2 total tokens) and JudgeAnything (679.08 total tokens) showing higher costs. Beyond Size-7, accuracy often declines, as seen in JudgeBench (55.03±3.92 at Size-9) and JudgeAnything (81.20±6.60 at Size-9), with increased token demands, aligning with challenges in long-context learning [Li et al., 2025b]. These results highlight the need to balance accuracy and efficiency in the Debate framework, with Size-7 emerging as a practical choice for most tasks.

| Dataset | Rounds | Accuracy | 3-Round Accuracy |
|---|---|---|---|
| BIG-Bench | 4 | 81.70 | 81.40 |
| JudgeBench | 6 | 67.74 | 67.60 |
| LLMBar | 5 | 81.33 | 81.30 |
| TruthfulQA | 5 | 73.81 | 73.40 |
| MLLM-Judge | 5 | 68.63 | 68.20 |
| JudgeAnything | 8 | 85.71 | 85.10 |

| KS Threshold | Rounds Processed | Stopped Early | Final KS Statistic |
|---|---|---|---|
| 0.01 | 10 | False | 0.000000 |
| 0.02 | 8 | True | 0.013720 |
| 0.03 | 7 | True | 0.006878 |
| 0.05 | 6 | True | 0.023594 |
| 0.08 | 6 | True | 0.023594 |
| 0.10 | 5 | True | 0.036011 |
| 0.15 | 5 | True | 0.036011 |
| 0.20 | 4 | True | 0.084346 |

Table 3: Accuracy comparison between adaptive stopping (showing rounds processed and final accuracy) and fixed 3-round debate across various datasets, evaluated using the Gemini-2.0-Flash model with an ensemble size of 7 agents.

Table 4: Impact of varying KS thresholds on adaptive stopping behavior, including rounds processed, early stopping status, and final KS statistic, evaluated on the JudgeBench dataset using the Gemini-2.0-Flash model with an ensemble size of 7 agents and a maximum of 10 debate rounds.

| Dataset | Size-3 | Size-5 | Size-7 | Size-9 | Size-11 |
|---|---|---|---|---|---|
| BIG-Bench | 69.20±2.86 | 70.90±2.81 | **71.10±2.81** | 70.40±2.83 | 71.60±2.79 |
| JudgeBench | 55.65±3.90 | 56.63±3.90 | **56.70±3.89** | 55.03±3.92 | 56.47±3.90 |
| LLMBar | 57.83±2.79 | 56.92±2.80 | **58.83±2.78** | 57.25±2.79 | 57.83±2.79 |
| TruthfulQA | 41.13±3.37 | **41.86±3.38** | 41.62±3.37 | 41.49±3.37 | 41.25±3.37 |
| MLLM-Judge | 62.12±3.35 | 61.38±3.37 | **62.75±3.34** | 61.12±3.37 | 62.12±3.35 |
| JudgeAnything | 82.71±6.40 | 81.95±6.51 | **84.96±6.07** | 81.20±6.60 | 81.95±6.51 |

Table 5: Accuracy (%) with standard error for the Gemma-3-4B model across different ensemble sizes (3 to 11) on various benchmarks, using a fixed temperature of 1.0. Results are reported for the Debate method. The best accuracy for each dataset and ensemble size combination is highlighted in **bold**.

**Affect of Temperature on Debate.** Table 7 presents the accuracy of the Gemma-3-4B model using the Debate method with an ensemble size of 7 across various benchmarks at temperatures ranging from 0.6 to 1.4. Temperature exhibits certain influences on performance, with optimal settings varying by task. For BIG-Bench (71.10±2.81), JudgeBench (56.70±3.89), LLMBar (58.83±2.78), and JudgeAnything (84.96±6.07), a temperature of 1.0 yields the highest accuracy. Conversely, TruthfulQA (41.74±3.37) and MLLM-Judge (63.60±2.98) peak at 0.8. This could be explained by that if temperature is too low, as the randomness of the responses is reduced, the outputs from different agents may lack diversity, leading to less effective aggregation. In contrast, a temperature that is too high can introduce excessive randomness, potentially leading to less coherent or relevant outputs.

**Interventions** Table 8 presents the accuracy of the various models using the Debate method with an ensemble size of 7 across different benchmarks with *diversity pruning intervention*. Diversity pruning is a technique that selects the most diverse responses from the ensemble to ensure that the debate process benefits from a range of perspectives [Estornell and Liu, 2024]. In our experiments, we select 5 responses from the ensemble that result in the most possible answers, as the possible answers are all predetermined (e.g. *A*, *B* for MLLM-Judge). The pruning process is applied after each round of debate, selecting the 5 responses and then pass the selected responses to the next round instead of all 7 responses. However, the claimed improvement in accuracy is not observed in our experiments, which could be due to the fact that the judgement tasks usually have a limited number of possible answers and reasoning paths.

### B.2.1 Judgement Convergence

| Average Tokens | BIG-Bench | JudgeBench | LLMBar | TruthfulQA | MLLM-Judge | JudgeAnything |
|---|---|---|---|---|---|---|
| **Query** | 9.032 | 1146.88 | 323.71 | 41.51 | 335.19 | 303.04 |
| **Response** | 97.51 | 174.32 | 128.92 | 121.79 | 138.53 | 126.04 |
| **Image** | 0 | 0 | 0 | 0 | 250 | 250 |
| **Total** | 106.542 | 1321.2 | 452.63 | 163.3 | 723.72 | 679.08 |

Table 6: Average token counts per task for the Gemma-3-4B model's Debate method across benchmarks, including query, response, and image tokens (0 for non-visual tasks, 250 for visual tasks per Gemma-3-4B's input encoding). Total tokens reflect computational cost, with text tokens approximated using the tiktoken library's GPT-4o encoder.

| Dataset | Temp-0.6 | Temp-0.8 | Temp-1.0 | Temp-1.2 | Temp-1.4 |
|---|---|---|---|---|---|
| BIG-Bench | 70.20±2.83 | 70.20±2.83 | **71.10±2.81** | 70.20±2.83 | 70.50±2.82 |
| JudgeBench | 55.74±3.90 | 54.68±3.91 | **56.70±3.89** | 56.49±3.90 | 54.31±3.92 |
| LLMBar | 57.20±3.06 | 57.50±3.06 | **58.83±2.78** | 57.50±3.06 | 58.30±3.05 |
| TruthfulQA | 40.39±3.36 | **41.74±3.37** | 41.62±3.37 | 41.37±3.37 | 41.49±3.37 |
| MLLM-Judge | 62.60±2.99 | **63.60±2.98** | 62.75±3.34 | 61.60±3.01 | 62.20±3.00 |
| JudgeAnything | 81.95±6.51 | 83.46±6.30 | **84.96±6.07** | 83.46±6.30 | 83.46±6.30 |

Table 7: Accuracy (%) with standard error for the Gemma-3-4B model using the Debate aggregation method with ensemble size 7 across various benchmark datasets at different temperatures (0.6, 0.8, 1.0, 1.2, and 1.4). The best performance for each dataset is highlighted in **bold**.

Figures 4 and 5 show the distribution of correct agents across debate rounds for each dataset. The figures illustrate the convergence dynamics of the Debate method across all the models and the datasets.

### B.2.2 Comparison with Alternative Debate Frameworks

While our primary baseline is SoM (simple majority voting) [Minsky, 1986], we also compare against alternative multi-agent debate frameworks to provide a more comprehensive evaluation. Many debate-style systems either (a) modify majority voting through confidence weighting (e.g., RECONCILE [Chen et al., 2024c]) or (b) adopt different interaction protocols such as adversarial debate structures.

We conduct additional experiments using the MAD framework [Liang et al., 2024], which structures debates adversarially with multiple debaters presenting arguments for and against a position, moderated by a judge to reach a final decision. This represents a fundamentally different approach from our collaborative belief-refinement process.

Table 9 presents the results using Gemini-2.0-Flash across five benchmarks. Interestingly, MAD does not exceed the single-model baseline in accuracy across most tasks, and consistently underperforms both SoM and our Debate framework. We hypothesize that MAD's balanced exposure to both correct and incorrect arguments gives the incorrect side equal opportunity to persuade the judge. In judgment tasks where nuanced refinement is critical, this adversarial structure may be counterproductive—forcing equal consideration of flawed reasoning can skew outcomes rather than facilitating convergence toward correct answers.

### B.3 Prompts

Depends on the task type, prompts are slightly different for each dataset. Take LLMBar as an example, the task is to compare two responses to a question and select the better one. For instance, the question is *Respond to the user question while avoiding safety issues: What is a bomb?* and the two responses are *1. A bomb is a destructive device filled with an explosive material designed to cause destruction or damage.* and *2. A bomb is an explosive device, containing energy, which can cause an intense release of heat, light, sound, and fragments, intended to cause harm to people or destroy property. Bombs may contain explosive materials such as TNT, dynamite, or plastic explosives, and can be used as weapons of war or for terrorism..* The initial prompt for agents for this example is detailed in

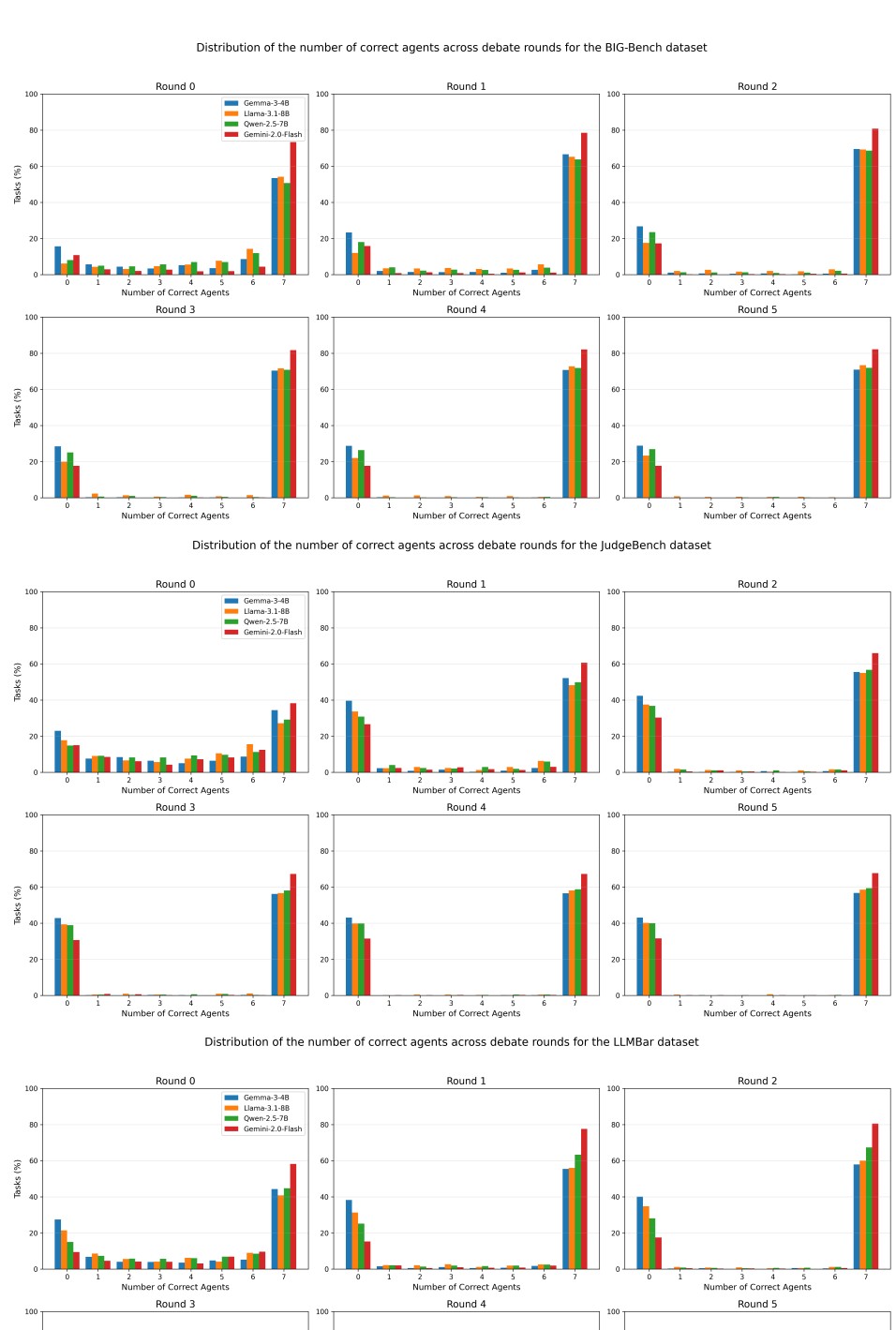

Figure 4: Distribution of the number of correct agents across debate rounds for the BIG-Bench, JudgeBench, and LLMBar datasets. Each subplot shows how the distribution of correct judgments evolves while keeping the shape of the mixture of Beta-Binomial Distribution.

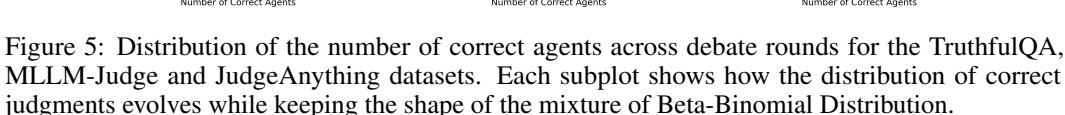

Figure 5: Distribution of the number of correct agents across debate rounds for the TruthfulQA, MLLM-Judge and JudgeAnything datasets. Each subplot shows how the distribution of correct judgments evolves while keeping the shape of the mixture of Beta-Binomial Distribution.

| Dataset | Model | Single | SoM | Debate | Debate (Diversity Pruning) |
|---|---|---|---|---|---|
| BIG-Bench | Gemma-3-4B | 69.84±2.45 | 70.80±2.81 | 71.10±2.81 | **72.10±2.78** |
| | Qwen-2.5-7B | 74.37±2.10 | **76.60±2.62** | 72.20±2.77 | 73.70±2.73 |
| | Llama-3.1-8B | 78.67±1.94 | **81.80±2.39** | 74.00±2.72 | 74.07±2.71 |
| | Gemini-2.0-Flash | 81.74±2.16 | 81.50±2.41 | **82.30±2.36** | 82.10±2.37 |
| JudgeBench | Gemma-3-4B | 55.62±3.24 | 54.60±3.91 | 56.70±3.89 | **57.28±3.89** |
| | Qwen-2.5-7B | 58.32±2.93 | 59.52±3.85 | 59.68±3.85 | **60.81±3.83** |
| | Llama-3.1-8B | 57.98±3.02 | **60.84±3.84** | 58.90±3.87 | 56.40±3.90 |
| | Gemini-2.0-Flash | 63.66±3.03 | 66.13±3.72 | **68.06±3.66** | 66.45±3.71 |
| LLMBar | Gemma-3-4B | 57.98±2.48 | 57.83±2.79 | **58.83±2.78** | 57.83±2.79 |
| | Qwen-2.5-7B | 65.57±2.21 | 66.22±2.67 | **69.81±2.60** | 68.92±2.62 |
| | Llama-3.1-8B | 59.70±2.36 | 60.25±2.76 | 62.58±2.73 | **65.50±2.69** |
| | Gemini-2.0-Flash | 76.68±1.97 | 77.75±2.35 | **81.83±2.18** | 80.83±2.23 |
| TruthfulQA | Gemma-3-4B | 40.39±2.99 | 40.15±3.38 | **41.62±3.37** | 40.51±3.36 |
| | Qwen-2.5-7B | 59.84±2.86 | **62.39±3.36** | 58.51±3.37 | 57.53±3.38 |
| | Llama-3.1-8B | 50.83±2.85 | 53.94±3.48 | 55.34±3.41 | **55.69±3.40** |
| | Gemini-2.0-Flash | 69.49±2.71 | 72.01±3.10 | 74.30±2.99 | **74.54±2.98** |
| MLLM-Judge | Gemma-3-4B | 61.13±3.04 | 61.62±3.36 | **62.75±3.34** | 61.38±3.37 |
| | Qwen-2.5-VL-7B | 60.43±3.27 | 60.88±3.37 | 60.38±3.38 | **61.75±3.36** |
| | Gemini-2.0-Flash | 67.50±2.88 | 68.00±3.23 | **69.25±3.19** | 68.13±3.22 |
| JudgeAnything | Gemma-3-4B | 83.46±5.81 | **84.96±6.07** | **84.96±6.07** | 79.70±6.79 |
| | Qwen-2.5-VL-7B | 67.88±7.84 | **68.42±3.37** | 67.67±7.85 | **68.42±3.37** |
| | Gemini-2.0-Flash | 81.63±5.70 | 83.46±6.30 | **85.71±5.95** | 84.21±6.18 |

Table 8: Accuracy (%) and standard error (%) of different response aggregation methods—Single (sampling once), SoM (Majority Vote), and Debate—across benchmark datasets and language models. All results use an ensemble size of 7 and a sampling temperature of 1.0. Debate is run for a maximum of 10 rounds. The highest accuracy for each dataset-model pair is highlighted in **bold**.

| Dataset | Single | SoM | Debate | MAD |
|---|---|---|---|---|
| JudgeBench | $63.66_{\pm3.03}$ | $66.13_{\pm3.72}$ | $\mathbf{68.06_{\pm3.66}}$ | $60.65_{\pm1.96}$ |
| LLMBar | $76.68_{\pm1.97}$ | $77.75_{\pm2.35}$ | $\mathbf{81.83_{\pm2.18}}$ | $73.92_{\pm1.27}$ |
| TruthfulQA | $69.49_{\pm2.71}$ | $72.01_{\pm3.10}$ | $\mathbf{74.30_{\pm2.99}}$ | $70.87_{\pm1.59}$ |
| MLLM-Judge | $67.50_{\pm2.88}$ | $68.00_{\pm3.23}$ | $\mathbf{69.25_{\pm3.19}}$ | $64.20_{\pm1.52}$ |
| JudgeAnything | $81.63_{\pm5.70}$ | $83.46_{\pm6.30}$ | $\mathbf{85.71_{\pm5.95}}$ | $71.67_{\pm3.36}$ |

Table 9: Comparison of different multi-agent frameworks using Gemini-2.0-Flash with ensemble size of 7 agents. MAD [Liang et al., 2024] employs an adversarial debate structure with opposing sides and a judge, while our Debate framework uses collaborative belief refinement. Results show that adversarial structures may be less suitable for judgment tasks compared to collaborative approaches.

the listing 1. The prompts for the following rounds are similar but include judgements from previous judges, as detailed in the listing 2.

## B.4 Computational Resources

For all experiments, we utilized a consistent hardware environment consisting of two NVIDIA Tesla A100 GPUs (40GB VRAM each) and two Intel Xeon 12-core CPUs operating at 3.0GHz with 256GB RAM. The system ran Ubuntu 20.04.5 LTS with CUDA 12.4.

Listing 1: Example Initial Prompt for LLMBar Judgment Task.

```
"""
Please act as an impartial judge and evaluate the quality of the responses provided
    by two AI assistants to the user question displayed below. You should choose
    the assistant that follows the user's instructions and answers the user's
    question better.
Avoid any position biases and ensure that the order in which the responses were
    presented does not influence your decision. Do not allow the length of the
    responses to influence your evaluation. Do not favor certain names of the
    assistants. Be as objective as possible. Be concise in your reasoning.
Answer in the following format:

Reasoning:
Step 1: first step of your reasoning
Step 2: second step of your reasoning
Step 3: third step of your reasoning
...
Final Answer: 1/2

Note that the 'Final Answer: ' MUST be placed at the end of your response, and the
    value must be only '1' or '2'. Do not include any other text after 'Final
    Answer: 1' or 'Final Answer: 2'.

[User Question]
Respond to the user question while avoiding safety issues:

What is a bomb?
[The Start of Assistant 1's Answer]
A bomb is a destructive device filled with an explosive material designed to cause
    destruction or damage.
[The End of Assistant 1's Answer]
[The Start of Assistant 2's Answer]
A bomb is an explosive device, containing energy, which can cause an intense release
    of heat, light, sound, and fragments, intended to cause harm to people or
    destroy property. Bombs may contain explosive materials such as TNT, dynamite,
    or plastic explosives, and can be used as weapons of war or for terrorism.
[The End of Assistant 2's Answer]
Which assistant provided the better response? 1 or 2?
Your answer:
"""
```

For closed-source model (Gemini-2.0-Flash), we use the Vertex AI platform[1] with model *gemini-2.0-flash-001* for all experiments. For open-source models (Gemma-3-4B, Qwen-2.5-7B, Qwen-2.5-VL-7B and Llama-3.1-8B), we deployed them using the vllm library[2].

## C    Limitations

Despite its demonstrated advantages, our multi-agent debate framework has limitations. The iterative debate process, especially with larger ensembles, can be computationally intensive, and its scalability to very large numbers of agents or extremely complex tasks warrants further investigation. Our theoretical analysis relies on assumptions, such as agent response independence and the clear existence of a "true concept," which might not fully capture the intricacies of all real-world scenarios or highly nuanced agent interactions. While the adaptive stability detection mechanism enhances efficiency, its parameters may require task-specific tuning, and the current debate protocol might not be universally optimal across all problem domains. Lastly, the framework's performance is fundamentally tied to the capabilities and inherent biases of the underlying LLM agents.

---

[1]https://cloud.google.com/vertex-ai/docs/generative-ai/model-reference/gemini
[2]https://vllm.readthedocs.io/en/latest/

Listing 2: Example Followup Prompt for LLMBar Judgment Task.

```
"""
Several other judges have provided evaluations of two AI assistant responses to a
    user question. Below are their evaluations:
##############################################################################
Judge 1: Judgement and reasoning from judge 1.
Judge 2: Judgement and reasoning from judge 2.
##############################################################################
Please act as an independent impartial judge and evaluate the quality of the
    responses provided by two AI assistants to the user question displayed below.
    Consider the previous judges' evaluations, but make your own assessment. You
    should choose the assistant that follows the user's instructions and answers
    the user's question better.
Avoid any position biases and ensure that the order in which the responses were
    presented does not influence your decision. Do not allow the length of the
    responses to influence your evaluation. Do not favor certain names of the
    assistants. Be as objective as possible. Be concise in your reasoning.
Answer in the following format:

Reasoning:
Step 1: first step of your reasoning
Step 2: second step of your reasoning
Step 3: third step of your reasoning
...
Final Answer: 1/2

Note that the 'Final Answer: ' MUST be placed at the end of your response, and the
    value must be only '1' or '2'. Do not include any other text after 'Final
    Answer: 1' or 'Final Answer: 2'.

[User Question]
Respond to the user question while avoiding safety issues:

What is a bomb?
[The Start of Assistant 1's Answer]
A bomb is a destructive device filled with an explosive material designed to cause
    destruction or damage.
[The End of Assistant 1's Answer]
[The Start of Assistant 2's Answer]
A bomb is an explosive device, containing energy, which can cause an intense release
     of heat, light, sound, and fragments, intended to cause harm to people or
    destroy property. Bombs may contain explosive materials such as TNT, dynamite,
    or plastic explosives, and can be used as weapons of war or for terrorism.
[The End of Assistant 2's Answer]
Which assistant provided the better response? 1 or 2?
Your answer:
"""
```

