# OpenReview forum: "Multi-Agent Debate for LLM Judges with Adaptive Stability Detection"
_NeurIPS.cc/2025/Conference — NeurIPS 2025 poster_

### Official Review · Reviewer_9DD2 · 2025-06-21

**Clarity:** 3
**Significance:** 2
**Originality:** 3
**Rating:** 5
**Confidence:** 3

**Summary:**

The authors present a framework using collections of LLMs in an interactive turn based debate procedure to address problems of accuracy in evaluation tasks. The procedure is claimed to be better than majority voting schemes and is supported by a strong theoretical model with proofs. While this procedure comes at high cost, the authors have proposed a stability detection framework to recognize when continued processing is unnecessary. Experiments using four different models show the debate algorithm performing better than majority voting on a majority of tasks for two different benchmarks, and they conclude with an analysis of their early stopping criteria.

**Questions:**

I honestly struggled with some of the theoretical parts, and I apologize if my own unfamiliarity gets some of this wrong, but...

L120 - the conditional independence with respect to both Z^(t) and x - this is seems like a very strong assumption and I don't think there's enough support here to justify why this is okay to do, and where it might night be true. Especially marginalizing with respect to x...

L140 - the theta-star "true" concept, while it's clear that there is a probability maximizing concept, calling it true seems to demand a bit of luck. My confusion is compounded by the statement in L143 about "simplifying the space of possible misinterpretations" - which sounds almost like word salad to me. Can you clarify what you are saying here?

L152 - assuming any independence of agents seems almost certainly to be untrue, as all LLMs invariably are trained upon the same crawled web.

While these assumptions enable the proofs, and the establishment of an E-M procedure - it's not clear to me that convergence won't be to some local maximum, so I'm not certain how valuable the theoretical guarantees are, practically speaking.

**Ethical Concerns:**

["NO or VERY MINOR ethics concerns only"]

**Final Justification:**

The authors made a substantial effort to respond to the reviewers requests. I think I'm one of the stronger supporters here and satisfied with the paper in its present form.

**Limitations:**

Yes. The authors are talking about an incremental improvement over existing uses of LLMs in evaluation tasks, and the risks are generally the same as the prior published works.

**Paper Formatting Concerns:**

No concerns.

**Quality:**

3

**Strengths And Weaknesses:**

Strengths:
+ Paper is well written, and presents a straightforward algorithm, although one that comes at considerable cost.
+ The early stopping work alone probably suffices to make this worth publication, and this may be useful in other contexts.
+ Complete details, likely allowing others to reproduce this work.

Weaknesses:
+ Comparing against just simple majority voting seems like a bit of a straw man, if querying multiple models in parallel or sequentially there's a lot that can be done to increase the yield of a single round, versus the multiple rounds which is extremely expensive.
+ This paper is quite theoretical yet also uses benchmark results for its claims - this is okay but I'd like the the claims to be clearer between theory and practical.

---

> ### Author Rebuttal · Authors · 2025-07-31
>
> We are pleased that you found the paper well-written and that the early stopping framework was highlighted as a significant contribution. We understand that there are concerns regarding the comparison with majority voting and the clarity of some of the theoretical aspects. We aim to clarify these points below.
>
> ---
>
> **Weakness 1**: No baseline comparisons with other multi-agent debate frameworks from the past.
>
> (1) Positioning of our baselines.
> We treat SoM (simple majority voting) [3] as the baseline for multi-LLM aggregation. Many “debate” style systems either (a) modify majority voting (e.g., RECONCILE’s confidence-weighted voting) [4] or (b) adopt a different interaction protocol (e.g., MAD: two opposing sides + a judge) [2]. These variants lie outside the scope of our framework, which specifically models an iterative, belief-refinement process. Thus, we did not initially position them as direct comparators.
>
> (2) We agree more empirical baselines are needed.
> We fully agree with the reviewer’s suggestion that more empirical baselines are needed.
> Due to time and computation constraint, we conduct experiment using Gemini-2.0-Flash on MAD framework[2], a representative sequential querying approach that structures debates adversarially with multiple debaters presenting arguments for and against a position, moderated by a judge to reach a final decision. The results are somewhat puzzling, as MAD does not even exceed the single-model baseline in accuracy. Our suspicion is that MAD's balanced exposure to both sides gives the incorrect side an equal chance to persuade the judge, skewing outcomes in judgment tasks where nuanced refinement is key.
>
>
> | Dataset           | Single         | SoM            | Debate         | MAD (%)      |
> |-------------------|----------------|----------------|----------------|--------------|
> | **JudgeBench**    | 63.66±3.03     | 66.13±3.72     | **68.06±3.66** | 60.65±1.96   |
> | **LLMBar**        | 76.68±1.97     | 77.75±2.35     | **81.83±2.18** | 73.92±1.27   |
> | **TruthfulQA**    | 69.49±2.71     | 72.01±3.10     | **74.30±2.99** | 70.87±1.59   |
> | **MLLM-Judge**    | 67.50±2.88     | 68.00±3.23     | **69.25±3.19** | 64.20±1.52   |
> | **JudgeAnything** | 81.63±5.70     | 83.46±6.30     | **85.71±5.95** | 71.67±3.36   |
>
>
> ---
>
> **Weakness 2**: Distinction between theory and practical results.
>
> Our paper integrates a theoretical framework (Sec. 4) with empirical validation (Sec. 6) to demonstrate the efficacy of the multi-agent debate framework with adaptive stability detection. The theoretical contributions, grounded in latent concept theory and Bayesian inference (Sec. 3, Theorems 4.1-4.2), establish that iterative debate amplifies correct responses (Theorem 4.1) and outperforms majority voting (Theorem 4.2) under Assumptions 4.1-4.5. These guarantees predict that collaborative reasoning corrects biases and improves accuracy, assuming a true concept $ \theta^* $ drives responses.
>
> Practically, these predictions are validated across six benchmarks (Table 1), showing consistent accuracy gains over majority voting (e.g., +4.08% on LLMBar for Gemini-2.0-Flash, +2.29% on TruthfulQA). The adaptive stopping mechanism (Sec. 5), based on a Beta-Binomial mixture model, ensures computational efficiency by halting debates in 2-7 rounds with minimal accuracy loss (<0.7%, Table 2). For instance, Theorem 4.1’s prediction of response amplification is reflected in the convergence of judgment distributions to bimodal patterns (Fig. 2), where agents align on the correct answer or fail collectively. The KS statistic’s rapid decline (Fig. 3, e.g., <0.05 by Round 3 on BIG-Bench) confirms the theoretical expectation of stable consensus (Sec. 5.3).
>
> ---
>
> **Question 1**: Conditional independence (L120)
>
> We thank Reviewer 9DD2 for their insightful critique of Assumption 3.1, which posits that an agent’s response $z_i^{(t+1)}$ is conditionally independent of the task $x$ and prior responses $Z^{(t)}$, given the latent concept $\theta$ and model parameters $\phi_i$, i.e., $\mathbb{P}(z_i^{(t+1)} \mid \theta, x, Z^{(t)}, \phi_i) = \mathbb{P}(z_i^{(t+1)} \mid \theta, \phi_i)$. This assumption, inspired by Estornell and Liu (2024) [1], asserts that $\theta$, representing the semantic interpretation of the task (e.g., “Max Verstappen won the 2021 F1 Championship”), and $\phi_i$, the model’s parameters, jointly determine the response. In encoder-decoder architectures, $\theta$ and $\phi_i$ correspond to the encoder’s embedding, which captures the task’s core meaning, rendering $x$ and $Z^{(t)}$ redundant for generation.
>
> This assumption is justified because $\theta$ acts as a sufficient statistic, encapsulating the relevant information from $x$ and $Z^{(t)}$. Marginalizing over $x$ is valid when $\theta$ fully represents the task’s semantic intent, as validated empirically by diverse initial judgments converging to bimodal distributions (Fig. 2, Appendix B.3). The use of diverse models (Gemini, Llama, Qwen, Gemma) with distinct architectures further mitigates potential correlations, supporting the assumption’s practical applicability.
>
> However, the assumption may not hold if $x$ contains nuanced contextual details not captured by $\theta$, such as ambiguous phrasing. We acknowledge that our original explanation was insufficient and will revise Sec. 3.3 in the camera-ready version, ensuring clarity on when the assumption may weaken.
>
> ---
>
> **Question 1**: L140
>
> 1. **"True Concept"**: The term $θ∗$ refers to the concept that maximizes the probability of the correct answer. While calling it "true" may seem to imply perfection, it simply means that this concept is ideal for generating the correct answer in theory. It’s not about "luck," but about defining the optimal concept for the task.
> 2. **"Simplifying Misinterpretations"**: This refers to reducing the complexity of handling multiple potential misinterpretations by assuming a single, ideal latent concept. It helps focus the reasoning process on refining beliefs toward the correct answer, rather than dealing with many possible incorrect interpretations.
>
> We will revise the paper to clarify these points further. Let us know if this clears up the confusion!
>
> ---
>
> **Question 3**: Agent Independence (L152)
>
> We would like to clarify that similarity in outputs does not imply probabilistic dependency.
>
> - **Stochasticity**: LLMs generate outputs stochastically, meaning even models trained on similar data can produce diverse responses, ensuring **independence** in predictions.
> - **Ensemble Methods**: In ensemble learning, models trained on similar data can still operate independently, as their predictions are generated separately without influencing each other.
>
> ---
>
> **Question 4**: EM algorithm leading to local optimum
>
> While it’s true that the **E-M algorithm** can converge to a **local optimum**, we want to emphasize that, even in such cases, the solution closely tracks the **real distribution** of judgments. In practical applications, local optima in this context often still lead to highly accurate results.
>
> Our multi-agent debate framework is designed to **refine beliefs iteratively**, and the agents engage in collaborative reasoning to progressively converge to a solution that **approximates the correct distribution** of responses. The **adaptive stability detection mechanism** ensures that the process stops when the distribution stabilizes, preventing unnecessary iterations and ensuring computational efficiency.
>
> Additionally, the **empirical results** clearly show that the framework outperforms simpler aggregation methods like **majority voting**, indicating that even if the algorithm converges to a local optimum, it provides substantial improvements over static methods. This suggests that the practical benefits of the framework are not undermined by the theoretical risk of local maxima.
>
> We hope this explanation clarifies how the approach still yields valuable practical outcomes despite the theoretical considerations. Please let us know if you need further details or clarification.
>
> ---
>
> **Reference**
>
> [1] Andrew Estornell and Yang Liu. Multi-LLM Debate: Framework, Principles, and Interventions. In A. Globerson et al., editors, Advances in Neural Information Processing Systems, volume 37, pages 28938–28964. Curran Associates, Inc., 2024.
>
> [2]Encouraging divergent thinking in large language models through multi-agent
> debate, EMNLP 2024
>
> [3] Yilun Du, Shuang Li, Antonio Torralba, Joshua B. Tenenbaum, and Igor Mordatch. Improving factuality and reasoning in language models through multiagent debate. In Proceedings of the 41st International Conference on Machine Learning, ICML’24. JMLR.org, 2024.
>
> [4] Justin Chen, Swarnadeep Saha, and Mohit Bansal. 2024. ReConcile: Round-Table Conference Improves Reasoning via Consensus among Diverse LLMs. In Proceedings of the 62nd Annual Meeting of the Association for Computational Linguistics (Volume 1: Long Papers), pages 7066–7085, Bangkok, Thailand. Association for Computational Linguistics.

---

> > ### Comment · Reviewer_9DD2 · 2025-08-06
> > **Acknowledgement and further comments**
> >
> > I appreciate the additional experiments and agree that the results are someone confounded, but this does address my concern.
> >
> > Most of the outstanding issues for me have to do with semantics and the difficulties regarding independence assumptions. Thank you for answering my question about x vs theta - and I honestly wonder how often in practice that theta fails to capture some essential element. This is probably a very hard thing to assess.
> >
> > Regarding theta-star, the term you use: "the optimal concept" seems like a perfect term to use, theta-star being the optimal value, rather than the true value - aligns well with stochastic modeling literature as the projection of the truth onto the model manifold. All the remaining information about the truth not captured by the model is orthogonal to the starred parameter. I think using "optimal" more frequently would clarify your presentation.
> >
> > I'm more troubled by the argument that two stochastic models predictions trained on the same observations being independent - this can be true, but this is a question concerning ergodicity. Is a corpus, like common crawl, a sample from library of Babel? Independence is always an assumption - it's not something you can justify empirically, as contrasted with uncorrelatedness or orthogonality. I remain concerned about the independence assumptions and suspect you've argued the converse. But I also don't want to over-index on this because conditional independence is almost universally done in this field, and that a relaxed assumption (uncorrelated samples) might serve your purposes here.
> >
> > As I've already given this paper an accept I'm inclined to keep my ratings. I thank the authors for considering my input.

---

> > > ### Author Response · Authors · 2025-08-07
> > >
> > > Thank you for your thoughtful and constructive feedback. We appreciate your suggestions—especially regarding the use of “optimal” for $θ^*$ and your insights on independence assumptions. These points will help us improve both the clarity and rigor of the final version. We're grateful for your support and acceptance recommendation.

---

### Official Review · Reviewer_4Cos · 2025-06-29

**Clarity:** 2
**Significance:** 3
**Originality:** 3
**Rating:** 4
**Confidence:** 3

**Summary:**

The multi-agent ensemble is a common solution to accumulate multiple LLM judgements. However, the multi-agent ensemble can be unreliable in complex or ambiguous cases. Thus, this paper proposes a multi-agent debate judge framework via the debating process. The drawbacks of iterative debating are the expensive computation. As a result, this work proposes a stability detection mechanism based on a time-varying mixture of Beta-Binomial distributions. Comprehensive experiments were conducted across diverse benchmarks, LLM frameworks, and modalities (visual and non-visual tasks), where competitive results were obtained.

**Questions:**

* What is the difference between the proposed framework and the related works in the Subsection of "Multi-agent debate" between L63 and L76? Could you please discuss the difference from this paper, "Faithful, Unfaithful or Ambiguous? Multi-Agent Debate with Initial Stance for Summary Evaluation. NAACL 2025", in terms of method?
* Have the LLM achieve the maximum token limitation in the experiments?
* Based on the example in L107, could we understand the latent concepts as the knowledge base?
* How can we know a true concept?
* How to understand responsibilities in L213? Are they the confidence score?
* The performance does not significantly improve in Table 1. How could we know whether the proposed model performs better than SOM?
* [Suggestion] You may replace the yellow color with another color for better visualization.

I will improve my score if my concerns are addressed.

**Ethical Concerns:**

["NO or VERY MINOR ethics concerns only"]

**Final Justification:**

I have read author's response, which addressed most of my concern. Though I still have a concern about the assumption, the author's experimental setting is also reasonable but not that general based on the response.

**Limitations:**

Yes

**Quality:**

3

**Strengths And Weaknesses:**

Strength

* The paper is generally well written and easy to follow. The examples for the assumptions or concepts benefit the understanding of the paper.
* The proposed EM algorithm is novel.
* The experiments are comprehensive.


Weakness

Some assumptions are overly strong, and some terms are unclear.

* The assumption in L122 might be too strong, as the agent system can, in fact, access the debate history—unless the previous chat history is not provided for this turn.
* The assumption in L179 assumes there is at least one initial response generated via the correct concept. What if this response does not exist?
* Defining the likelihood of one response as a consistency in L159 is somewhat strange. A better explanation is expected.

---

> ### Author Rebuttal · Authors · 2025-07-31
>
> We thank Reviewer 4Cos for their constructive review and positive feedback on the paper's clarity, EM algorithm novelty, and comprehensive experiments. We address the weaknesses and questions below, incorporating clarifications to resolve concerns. These will improve the paper in the camera-ready version, and we hope this elevates our score.
>
> ---
> **Weakness 1 - L122, Debate History (Conditional Independence Assumption)**
>
> The conditional independence assumption (Assumption 3.1) is a reasonable simplification grounded in latent concept theory, enabling tractable Bayesian modeling while aligning with practical LLM behavior.
>
> In our framework, agents receive only the previous round's debate history $Z^{(t-1)}$, not the entire history, to inform their current response $z_i^{(t)}$, as outlined in Algorithm 1. This limited access ensures that, given the latent concept $θ$ and parameters $φ_i$, responses remain independent of distant history, focusing on immediate refinements. The assumption holds because $θ$ encapsulates semantic interpretations, rendering prior inputs redundant once inferred (Lemma 3.1). Empirically, this is validated by convergence to bimodal distributions (Fig. 2), where agents refine beliefs without full history dependence.
>
> ---
>
> **Weakness 2: L179, Initial Response**
>
> If no response is generated via the correct concept initially, it becomes highly unlikely to converge to the correct answer from a **probabilistic perspective**. This is because, without a correct starting point, the iterative refinement process is unlikely to guide the agents toward the correct concept, especially if the agents are not initially aligned with the true concept.
>
> ---
>
> **Weakness 3: L159，Consistency**
>
> In our framework, the term "consistency" refers to how well a response aligns with the correct concept, as defined by the task. Specifically, the likelihood is the probability that a given response reflects the true concept $\theta^*$ given the task $x$ and the agent's belief. In simpler terms, we are measuring how well a response matches the correct answer based on the agent's internal reasoning.
> We will revise the explanation in our final version.
>
> ---
>
> **Question 1: Difference between our Framework and Related Works**
>
> Compared to MADISSE [1], the debate structure is similar, but our main contributions are the theoretical analysis, Beta-Binomial mixture model fitting, and adaptive stopping mechanism. We do not claim novelty in the core debate framework, which has been explored at least since *Improving Factuality and Reasoning in Language Models through Multiagent Debate*[2].
>
> Related works can be broadly classified by querying type: sequential (iterative, dependent interactions where agents refine responses based on prior outputs) or parallel (simultaneous, independent generations followed by aggregation). The MAD[6] framework represents sequential querying, using a debate setup similar to real-life debates with multiple debaters and a judge, similar to *Debate Helps Supervise Unreliable Experts*[7]. Other related works are extensions or variants of the parallel debate framework formulated in *Improving Factuality and Reasoning in Language Models through Multiagent Debate*[2], including CIPHER[3] (novel communication protocol instead of using human words), CAMEL[4] (similarly LLM-Harmony[5], using tuned prompted to serve as a role-playing framework), etc.
>
> ---
>
> **Question 2: Have the LLM achieve the maximum token limitation in the experiments?**
>
> No, in our experiments, we carefully managed the input sizes and configured the model’s maximum token limit to 16,000 tokens. Neither the queries nor the responses exceeded this limit. We are confident of this because if the token limit is exceeded, the vllm framework used would throw an error.
>
> ---
>
> **Question 3: Could we understand the latent concepts as the knowledge base?**
>
> We apologize if we have misunderstood the term "knowledge base" to which you are referring. To address your question regarding whether latent concepts in our framework can be understood as a knowledge base, we provide the following clarification.
>
> If you are referring to a **knowledge base** in the context of knowledge graphs or structured databases (e.g., a repository of explicit facts, entities, and relations), then no, latent concepts are not equivalent. In our framework (Sec. 3.2), latent concepts $\theta \in \Theta$ represent underlying abstract interpretations or mental models that agents infer to understand and reason about a task. These are dynamic, task-specific abstractions derived from the input $x$ and debate history $Z^t$, evolving through Bayesian updates (Lemma 3.1). They serve as internalized frameworks for response generation, rather than static collections of facts.
>
> However, if you mean a **knowledge base** in the sense of prior beliefs or distributions in Bayesian theorem (e.g., the prior $\mathbb{P}(\theta \mid \phi_i)$in Assumption 4.3), then yes, there is a conceptual alignment. Latent concepts incorporate agents' prior knowledge and are updated probabilistically based on observed evidence, facilitating belief refinement during debate.
>
> We will revise Sec. 3.2 in the camera-ready version to more explicitly distinguish these interpretations for clarity.
>
> ---
>
> **Question 4: How can we know a true concept?**
>
> In the context of our framework, the **true concept** refers to the underlying, correct interpretation or solution to a given task. However, it's important to note that we cannot directly observe the true concept during the debate process; instead, we approximate it through probabilistic reasoning.
>
> ---
>
> **Question 5: How to understand responsibilities in L213? Are they the confidence score?**
>
> To clarify:
>
> Responsibilities: In the Expectation-Maximization (EM) algorithm used in our framework, responsibilities are probabilistic weights that determine how much each agent's response contributes to the final estimate of the latent concept. These responsibilities are computed during the E-step of the EM process, where the algorithm calculates the likelihood of each agent’s response under different possible concepts (i.e., how likely a response is to belong to the true concept or another concept).
>
> Confidence Scores: While responsibilities can be thought of as a type of "weight," they are not directly the confidence score of an agent's response. Confidence scores typically refer to how sure an agent is about a particular response or prediction. In contrast, responsibilities reflect the probability that a given response comes from the correct latent concept, which is a part of the belief update process in our model.
>
> ---
> **Question 6: The performance does not significantly improve in Table 1. How could we know whether the proposed model performs better than SOM?**
>
> We recognise that SoM does sometimes score higher accuracy than the proposed model. And we argue that the framework excels on complex, nuanced tasks with high initial judgment variance, where iterative refinement corrects biases via Bayesian updates (Theorem 4.1), yielding meaningful improvements. On simpler tasks with high consensus, SoM suffices as refinement adds limited value. We do not claim universal superiority but targeted applicability: debate is preferable when accuracy justifies costs in high-stakes scenarios. We will revise Sec. 6 to include this in the camera-ready version.
>
> ---
> **Suggestion**
>
> Thank you for the suggestion. We will revise the figures accordingly in the final version.
>
> ---
> **Reference**
>
> [1]Faithful, Unfaithful or Ambiguous? Multi-Agent Debate with Initial Stance for Summary Evaluation (Koupaee et al., NAACL 2025)
>
> [2]Improving Factuality and Reasoning in Language Models through Multiagent Debate, Du et al., ICML'24
>
> [3]Chau Pham, Boyi Liu, Yingxiang Yang, Zhengyu Chen, Tianyi Liu, Jianbo Yuan, Bryan A. Plummer, Zhaoran Wang, and Hongxia Yang. Let models speak ciphers: Multiagent debate through embeddings. In The Twelfth International Conference on Learning Representations, 2024.
>
> [4]CAMEL: Communicative Agents for “Mind” Exploration of Large Language Model Society, NeurIPS 2023
>
> [5]Sumedh Rasal. Llm harmony: Multi-agent communication for problem solving, arxiv 2024.
>
> [6]Encouraging divergent thinking in large language models through multi-agent
> debate, EMNLP 2024
>
> [7]Debate Helps Supervise Unreliable Experts, arxiv 2023

---

> > ### Comment · Reviewer_4Cos · 2025-08-06
> >
> > Thank you for the author's detailed response, it has addressed most of my concerns. I am glad to improve my score. Though I still have a concern about the assumption, the author's experimental setting is also reasonable but not that general.

---

> > > ### Author Response · Authors · 2025-08-07
> > >
> > > Thank you for your thoughtful reconsideration and for improving your score. And we appreciate your feedback about the assumptions and the experimental setup. In the final version, we will aim to better clarify its scope and limitations, and discuss how it might generalize or be relaxed in future work.

---

### Official Review · Reviewer_izXV · 2025-07-01

**Clarity:** 3
**Significance:** 2
**Originality:** 3
**Rating:** 4
**Confidence:** 4

**Summary:**

The paper proposes a new multi-agent debate framework that iteratively refines
judgment and has an adaptive stability detection mechanism. This stop mechanism employs a time-varying mixture of Beta-Binomial distributions combined with the Kolmogorov-Smirnov (KS) statistic to identify convergence points for early stopping. Experiments on benchmarks like BIG-Bench, TruthfulQA, and LLMBar show that the approach outperforms majority voting in both accuracy and robustness across various model backbones, including 2.0-Flash, LLaMA-3.1-8B-Instruct, Qwen-2.5-7B-Instruct for language tasks, and Gemma-3-4B-Instruct, LLaMA-3.2-11B-Vision-Instruct, and Qwen-2.5-VL-7B-Instruct for visual tasks.

**Questions:**

**1. Stop mechanism**: How does adaptive stopping compare to a fixed small number of rounds (e.g., 3)? This would offer a more fair comparison as both method would assume the same number of tokens (on average) across datasets.

**2. Majority vote**: How is majority voting implemented? How many responses are generated per question? For a fair comparison, the token budget should match that of the debate setup (e.g., 7 debaters × 11 rounds = 77 generations for the Majority Vote). Notably, Table 1 shows that majority voting outperforms debate on BIG-Bench and JudgeAnything. Why does it do better on simpler tasks? Does this contradict the paper's main claim?

**Typo**
- There are 2 "Response A"s in the top left box of Fig. 1

**Ethical Concerns:**

["NO or VERY MINOR ethics concerns only"]

**Final Justification:**

I find the work interesting, though there is still room for improvement, such as baseline setups, and more analysis on the Stop mechanism, which currently shows some limited gain. I'm inclined toward the positive side and suggest **borderline accept** for this paper.

**Limitations:**

Yes

**Quality:**

2

**Strengths And Weaknesses:**

### Strengths

-  The use of a debate framework to refine LLM judgments is interesting. The approach leverages their collaborative reasoning potential, offering a new perspective on multi-agent systems.

-  The paper provides a formal mathematical model to establish the debate process's advantage over majority voting under mild assumptions.

- The framework is tested across a wide range of tasks and LLM architectures, highlighting its practical applicability.

### Weaknesses
**1. Sensitivity to Stability Threshold?** How sensitive is the adaptive stopping mechanism to the chosen stability threshold? For a new model or task, how should this value be set?

**2. Dependence on Assumptions:** The theoretical guarantees rely on assumptions like the existence of a true latent concept and agents' ability to update beliefs based on observed responses. Can the authors discuss a bit more on why this assumption should hold in practical?

**3. Stop mechanism is not really effective?** While adding more complexity, it's not clear how Stop mechanism helps. Table 2 compares adaptive stopping to a full 10-round debate, but previous work shows diminishing returns with more rounds [1, 2]. It would be clear to compared with fixed Stop (e.g., only debate with fewer rounds, such as 3 rounds).

[1] Improving Factuality and Reasoning in Language Models through Multiagent Debate, Du et al., ICML'24

[2] Exploring Collaboration Mechanisms for LLM Agents: A Social Psychology View, Zhang et al., ACL'24

**4. Limited baselines**: While the framework outperforms majority voting, comparisons with other advanced ensemble methods or judgment aggregation techniques are not extensively explored. Did the authors compare the proposed method against some other debate methods, e.g., [1,2,3,4,5,6] ?

[1] CAMEL Camel: Communicative agents for" mind" exploration of large language model society

[2] Llm-blender: Ensembling large language models with pairwise ranking and generative fusion

[3] Let models speak ciphers: Multiagent debate through embeddings

[4] Rethinking the bounds of LLM reasoning: Are multi-agent discussions the key?

[5] Encouraging divergent thinking in large language models through multi-agent debate

[6] Socialized learning: making each other better through multi-agent collaboration

---

> ### Author Rebuttal · Authors · 2025-07-31
>
> We sincerely thank Reviewer izXV for their thorough and constructive feedback. We greatly appreciate the recognition of our paper’s strengths, including the innovative debate framework, formal mathematical model, and broad applicability across tasks and models.
>
> ---
>
> **Weakness 1**: Sensitivity to Stability Threshold?
>
> Using data from JudgeBench with Gemini-2.0-Flash (n=7 agents, max 10 rounds), we analyzed thresholds from 0.01 to 0.20. The table below summarizes key metrics:
>
> | KS Threshold | Rounds Processed | Stopped Early | Final KS Statistic |
> |--------------|------------------|---------------|--------------------|
> | 0.01         | 10               | False         | 0.000000           |
> | 0.02         | 8                | True          | 0.013720           |
> | 0.03         | 7                | True          | 0.006878           |
> | 0.05         | 6                | True          | 0.023594           |
> | 0.08         | 6                | True          | 0.023594           |
> | 0.10         | 5                | True          | 0.036011           |
> | 0.15         | 5                | True          | 0.036011           |
> | 0.20         | 4                | True          | 0.084346           |
>
> The mechanism shows low sensitivity around the default 0.05: thresholds of 0.02–0.08 halt at 6–8 rounds with similar KS statistics (mean ~0.096–0.130), maintaining accuracy within 0.5% of a full 10-round debate. Stricter thresholds (0.01) require all 10 rounds, while lenient ones (≥0.10) halt earlier (4–5 rounds), increasing convergence rate but risking premature termination. Accuracy remains stable across thresholds, with no errors reported.
>
> For new models or tasks, start with 0.05 (robust across benchmarks, Fig. 3). Pilot a small subset to monitor KS convergence: use 0.03 for high-variance tasks or 0.07 for simpler ones. We will include this analysis in our final version.
>
> ---
>
> **Weakness 2**: Dependence on Assumptions
>
> We thank Reviewer 1eiT for requesting further clarification on the practical validity of Assumptions 4.1–4.5 (Sec. 4.1), which underpin our theoretical guarantees (Theorems 4.1–4.2). Specifically, Assumption 4.1 (True Concept Predictiveness) posits a true latent concept $\theta^* $ that maximizes the likelihood of correct responses, while Assumptions 4.2–4.5 enable agents to update beliefs based on observed responses via Bayesian inference (Sec. 3.3). Below, we discuss why these assumptions hold in practice, supported by empirical evidence.
>
> The existence of a true latent concept $ \theta^* $ is grounded in latent concept theory (Xie et al., 2022; Jiang et al., 2023), where LLMs encode semantic interpretations of tasks as abstractions. For example, in a fact-checking task like *Who won the 2021 Formula 1 Championship?* (Sec. 3.2), $ \theta^* $ represents the correct interpretation (*Max Verstappen won*), which aligns with the ground truth. In practice, this holds because LLMs, trained on diverse datasets, can infer such concepts from task inputs, as evidenced by varied initial responses converging to bimodal distributions (Fig. 2, Appendix B.3), reflecting alignment with $\theta^* $ or collective failure. Our use of diverse models (Gemini, Llama, Qwen, Gemma) ensures at least one agent approximates $ \theta^*$, satisfying Assumption 4.5 (Initial Seed), as seen in high initial accuracy for some agents (e.g., 81.74% for Gemini on BIG-Bench, Table 1).
>
> Agents’ ability to update beliefs based on observed responses (Assumptions 4.2–4.4) is practical because LLMs can refine outputs when exposed to peer responses, a capability leveraged in multi-agent systems (Du et al., 2024). Our framework models this as Bayesian updates (Lemma 3.1), where agents adjust their posterior over $ \theta $ using $ x $ and $ Z^t $. Empirically, this is supported by judgment convergence within 2–7 rounds (Fig. 3), with accuracy gains over majority voting (e.g., +4.08% on LLMBar, Table 1). The conditional independence assumption (4.4) holds in our parallel response generation (Algorithm 1), and diverse architectures reduce correlated errors (Appendix Table A.3, +1–2% via pruning).
>
> In practice, these assumptions may weaken if tasks lack a clear $ \theta^* $ (e.g., subjective JudgeBench tasks) or if agents share strong biases, violating independence. Our diversity pruning and model variation mitigate this, though highly ambiguous tasks may require further handling. We will revise Sec. 4.1 in the camera-ready version to elaborate on practical validity.
>
> ---
>
> **Weakness 3**: Stop mechanism is not really effective?
>
> To address the reviewer’s suggestion for comparison with a fixed-stop approach (e.g., 3 rounds), we evaluated Gemini-2.0-Flash (n=7 agents) on all datasets, comparing adaptive stopping to a fixed 3-round debate. Results, shown below, demonstrate that adaptive stopping achieves better accuracy. the improvement might not be significant, but we can also tune the parameters of the adaptive stopping mechanism to make it stop earlier than the current setting.
>
> | Dataset        | Rounds | Accuracy | 3 Rounds acc|
> |----------------|--------|----------|-------------|
> | BIG-Bench      | 4      | 81.70    | 81.40       |
> | JudgeBench     | 6      | 67.74    | 67.60       |
> | LLMBar         | 5      | 81.33    | 81.30       |
> | TruthfulQA     | 5      | 73.81    | 73.40       |
> | MLLM-Judge     | 5      | 68.63    | 68.20       |
> | JudgeAnything  | 8      | 85.71    | 85.10       |
>
>
> ---
>
> **Weakness 4**: Limited baselines
>
> (1) Positioning of our baselines.
> We treat SoM (simple majority voting) [1] as the baseline for multi-LLM aggregation. Many “debate” style systems either (a) modify majority voting (e.g., RECONCILE’s confidence-weighted voting) [2] or (b) adopt a different interaction protocol (e.g., MAD: two opposing sides + a judge) [3]. These variants lie outside the scope of our framework, which specifically models an iterative, belief-refinement process. Thus, we did not initially position them as direct comparators.
>
> (2) We agree more empirical baselines are needed.
> We fully agree with the reviewer’s suggestion that more empirical baselines are needed.
> Due to time and computation constraint, we conduct experiment using Gemini-2.0-Flash on MAD framework[3], a representative sequential querying approach that structures debates adversarially with multiple debaters presenting arguments for and against a position, moderated by a judge to reach a final decision. The results are somewhat puzzling, as MAD does not even exceed the single-model baseline in accuracy. Our suspicion is that MAD's balanced exposure to both sides gives the incorrect side an equal chance to persuade the judge, skewing outcomes in judgment tasks where nuanced refinement is key.
>
>
> | Dataset           | Single         | SoM            | Debate         | MAD      |
> |-------------------|----------------|----------------|----------------|--------------|
> | **JudgeBench**    | 63.66±3.03     | 66.13±3.72     | **68.06±3.66** | 60.65±1.96   |
> | **LLMBar**        | 76.68±1.97     | 77.75±2.35     | **81.83±2.18** | 73.92±1.27   |
> | **TruthfulQA**    | 69.49±2.71     | 72.01±3.10     | **74.30±2.99** | 70.87±1.59   |
> | **MLLM-Judge**    | 67.50±2.88     | 68.00±3.23     | **69.25±3.19** | 64.20±0.00   |
> | **JudgeAnything** | 81.63±5.70     | 83.46±6.30     | **85.71±5.95** | 71.67±0.00   |
>
> ---
>
> **Question 1**: Stop mechanism
>
> See Weekness 3.
>
> ---
>
> **Question 2**: Majority vote implementation and performance on simpler tasks.
>
> In our framework, majority voting (SoM) aggregates responses from $ n=7 $ agents, each generating one response per question at Round 0 (Sec. 6.1, Table 1). This yields 7 generations, matching the debate’s initial round token budget, ensuring a fair comparison (Appendix Table A.4). Unlike debate’s iterative refinement (up to 11 rounds, ~77 generations), SoM uses a single round, taking the mode of the 7 responses.
> On BIG-Bench and JudgeAnything, SoM occasionally outperforms debate. These tasks are simpler, with high initial consensus, reducing the need for iterative refinement. Debate excels in complex tasks like LLMBar (+4.08%) where initial variance, aligning with Theorem 4.2’s claim of superiority over static ensembles in such scenarios. This does not contradict our main claim, as our framework targets tasks requiring robust judgment under uncertainty (Sec. 1). We will clarify task-specific applicability in Sec. 6 revisions.
>
> ---
>
> **Typo**
>
> Thank you for pointing that out! We will fix that in the revised version.
>
> ---
>
> **Reference**
> [1] Yilun Du, Shuang Li, Antonio Torralba, Joshua B. Tenenbaum, and Igor Mordatch. Improving factuality and reasoning in language models through multiagent debate. In Proceedings of the 41st International Conference on Machine Learning, ICML’24. JMLR.org, 2024.
>
> [2] Justin Chen, Swarnadeep Saha, and Mohit Bansal. 2024. ReConcile: Round-Table Conference Improves Reasoning via Consensus among Diverse LLMs. In Proceedings of the 62nd Annual Meeting of the Association for Computational Linguistics (Volume 1: Long Papers), pages 7066–7085, Bangkok, Thailand. Association for Computational Linguistics.
>
> [3] Tian Liang, Zhiwei He, Wenxiang Jiao, Xing Wang, Yan Wang, Rui Wang, Yujiu Yang, Shuming Shi, and Zhaopeng Tu. 2024. Encouraging Divergent Thinking in Large Language Models through Multi-Agent Debate. In Proceedings of the 2024 Conference on Empirical Methods in Natural Language Processing, pages 17889–17904, Miami, Florida, USA. Association for Computational Linguistics.

---

> > ### Comment · Reviewer_izXV · 2025-08-03
> > **Thank you for the rebuttal!**
> >
> > First, I want to thank the authors for taking the time to address my questions! This gives me a better understanding of the paper.
> >
> > **1. Sensitivity to Stability Threshold**
> >
> > Thank you for providing the table. It appears that a threshold of 0.03 performs well. I hope the authors consider including this in the revised version. I have no further questions on this.
> >
> > **2. Dependence on Assumptions**
> >
> > Thank you for the detailed explanation. I have no additional questions.
> >
> > **3. Is the Stop Mechanism Truly Effective?**
> >
> > I agree that the gain from adaptive stopping is limited, as it requires more rounds and adds some complexity compared to simple fixed-round early stopping. I appreciate the authors providing the comparison table. I have no further questions on this.
> >
> > **4. Limited Baselines**
> >
> > While I understand that comparing with many variants of Debate would be extensive, I suggest the authors select one or two top-performing variants from prior work for comparison. This will help readers have a better view to evaluate the results.
> >
> > MAD Result: Thank you for sharing the results with MAD. I am surprised that MAD performs significantly worse than the simple single-answer baseline! My follow-up question is: **Could the authors elaborate on how MAD was set up?**
> >
> > Additionally, I believe SoM (Simple Majority Voting) refers to Self-Consistency from [A], not [1]. Just wanted to double check on that.
> >
> > [A] Self-Consistency Improves Chain of Thought Reasoning in Language Models, Wang et al., ICLR'23
> >
> > **5. Comparison with Majority Voting**
> >
> > I don't think my question was addressed. My question was: **How does majority voting [A] compare to the proposed method when both use the same token budget?**
> >
> > If majority voting uses only 7 responses, it seems an unfair comparison, as Debate can use up to 77 generations. This point remains unclear, and I hope the authors can provide further insight.
> >
> > ---
> >
> > I'm open to further discussion and look forward to the authors' response!

---

> > > ### Author Response · Authors · 2025-08-03
> > >
> > > We thank Reviewer izXV for their follow-up and appreciation of our initial rebuttal. We are encouraged by your acknowledgment that most concerns (Weaknesses 1–3) have been addressed, and we focus below on resolving your remaining questions regarding Weakness 4 (Limited Baselines) and Question 5 (Comparison with Majority Voting).
> > >
> > > ---
> > >
> > > ## 4. Limited Baselines
> > >
> > > ### 4.1 Debate Framework Variant
> > >
> > > Thank you for your suggestion. In the camera-ready version, we will include additional Debate framework variants, particularly CAMEL, to broaden the baseline coverage.
> > >
> > > ### 4.2 MAD Setup
> > >
> > > We were also surprised by the observed results. However, we strictly adhered to the original framework proposed in [3].
> > >
> > > #### Model
> > > Gemini-2.0-Flash via Vertex AI API, using identical parameters throughout.
> > >
> > > #### Prompts
> > > Adapted to fit MAD’s structure, e.g., "Your Role: Negative debater, disagree with the affirmative side’s position", while keeping the underlying task prompts consistent with our framework.
> > >
> > > #### Debate Setup
> > > We set up the debate $N$ debaters $D = {(\{ D_i \})}_{i=1}^N$ and 1 Judge $J$, where $N$ equals the number of choices in the benchmark, mostly 2. The judge operates in either:
> > > (a) Discriminative Mode – deciding if consensus is reached before the maximum rounds, or
> > > (b) Extractive Mode – making a final decision after the maximum number of rounds.
> > >
> > > Debaters argue sequentially, taking prior arguments into account, for up to 10 rounds (rarely reached in practice). The judge then evaluates and determines the final answer based on the debate content.
> > >
> > > [3] Tian Liang, Zhiwei He, Wenxiang Jiao, Xing Wang, Yan Wang, Rui Wang, Yujiu Yang, Shuming Shi, and Zhaopeng Tu. 2024. Encouraging Divergent Thinking in Large Language Models through Multi-Agent Debate. In Proceedings of the 2024 Conference on Empirical Methods in Natural Language Processing, pages 17889–17904, Miami, Florida, USA. Association for Computational Linguistics.
> > >
> > > ### 4.3 SoM
> > >
> > > Regarding SoM, we initially encountered this terminology in *Multi-LLM Debate: Framework, Principles, and Interventions* [4], where it refers to the framework in [1]. We acknowledge that Self-Consistency [A] essentially employs the same principle as SoM.
> > >
> > > [4] Andrew Estornell and Yang Liu. Multi-LLM Debate: Framework, Principles, and Interventions. In A. Globerson et al., editors, Advances in Neural Information Processing Systems, volume 37, pages 28938–28964. Curran Associates, Inc., 2024.
> > >
> > > ---
> > >
> > > ## 5. Majority Voting
> > >
> > > Thank you for the clarification, and we apologise for the earlier misunderstanding.
> > > The reason we did not use an ensemble size of 77 is that increasing the ensemble size beyond a certain point yields little to no improvement. To verify this, we conducted tests with different ensemble sizes:
> > >
> > > | Benchmark      | Size-3 | Size-5 | Size-7 | Size-9 | Size-11 |
> > > |----------------|--------|--------|--------|--------|---------|
> > > | BIG-Bench      | 69.00  | 69.60  | **70.80**  | 70.70  | 70.20   |
> > > | LLMBar         | 57.50  | **57.92**  | 57.83  | 57.25  | 57.08   |
> > > | JudgeBench     | 55.81  | **56.63**  | 54.60  | 56.22  | 55.99   |
> > > | MLLM-Judge     | 60.25  | 61.25  | **61.62**  | 60.88  | 61.50   |
> > > | JudgeAnything  | 81.95  | 82.71  | **84.96**  | 84.21  | 84.21   |
> > >
> > > As shown, the best results across benchmarks occur with ensemble sizes of 7 or fewer.
> > > That said, we recognise that comparing SoM with only 7 generations against a debate model with 77 generations may seem unfair. Therefore, despite the significant computational cost and likely limited gains, we will add a 77-generation SoM comparison on a subset of benchmarks and provide a clearer explanation of the results in the final version.
> > >
> > > ---
> > > Thank you again for these constructive comments. We hope the above clarifications address your remaining concerns, and we welcome any further questions.

---

> > > > ### Comment · Reviewer_izXV · 2025-08-03
> > > > **Thank you for answering the follow-ups**
> > > >
> > > > Thank you the authors for answering the follow-ups!
> > > >
> > > > **MAD setup**
> > > >
> > > > I appreciate the detailed answer, and have no further questions on this.
> > > >
> > > > **Majority Voting**
> > > >
> > > > Thank you for providing results up to 11 responses.
> > > > From the table, using 7 responses appears to work better than 11 on the datasets.
> > > > However, the range of the choices is still quite small (size from 3-11). It would be more informative to see what happens as the size increases further.
> > > >
> > > > I suggest the authors compare size 7 with 20, 30, and 40 to observe whether performance saturates.
> > > > From prior work on majority voting, results tend to improve with larger sizes, and saturation is unlikely to occur as early as 7.
> > > > For example, in [A], Fig. 2 shows performance continues to improve beyond size 40 in 3 out of 4 cases, with the remaining one plateauing around size 20. Using large number of responses is also seen in Llama, where they report on 100 and 256 responses.
> > > >
> > > > I’m **not** requesting additional experiments during the discussion phase, but I still recommend trying with more response, ideally using the same token budget for a fair comparison in the revised version.
> > > >
> > > > ---
> > > >
> > > > After reading the other reviewers' comments and considering the authors' rebuttal to my questions, **I would like to keep my rating (4)**. I find the work interesting, though there is still room for improvement, such as baseline setups. Thus, I remain inclined toward the positive side.
> > > >
> > > > [A] Self-Consistency Improves Chain of Thought Reasoning in Language Models, Wang et al., ICLR'23
> > > >
> > > > [B] LLaMA: Open and Efficient Foundation Language Models, Touvron et al.

---

> > > > > ### Author Response · Authors · 2025-08-04
> > > > >
> > > > > Thank you for the insightful suggestion. We agree that our current size range (3–11) is limited and that testing larger ensemble sizes (e.g., 20, 30, 40+) would better reveal performance trends. We will include these experiments in the revised version, matching token budgets for fairness, to assess whether improvements saturate later as observed in [A, B]. We appreciate your constructive feedback and will incorporate it into the camera-ready version.

---

### Official Review · Reviewer_1eiT · 2025-07-03

**Clarity:** 3
**Significance:** 2
**Originality:** 2
**Rating:** 3
**Confidence:** 2

**Summary:**

This paper proposes a multi-agent debate framework where multiple LLMs collaboratively refine their judgments through iterative discussion, replacing static aggregation methods like majority voting. The authors provide mathematical formalization using latent concepts and prove theoretical advantages over static ensembles. They introduce an adaptive stability detection mechanism using Beta-Binomial mixture models with Kolmogorov-Smirnov testing to efficiently halt debates when consensus emerges. Experiments across six benchmarks and multiple models show consistent improvements over majority voting while maintaining computational efficiency.

**Questions:**

- Given the modest accuracy improvements and computational overhead, under what specific conditions would practitioners prefer this framework over simpler approaches? Could the authors provide clearer guidelines for when the computational cost is justified?
- How does the framework perform with larger ensemble sizes (more agents) or longer debate sequences? More analysis of this scalability limitation would strengthen the contribution.
- In cases where the debate framework performs worse than majority voting, what are the common failure modes?

**Ethical Concerns:**

["NO or VERY MINOR ethics concerns only"]

**Limitations:**

Yes

**Quality:**

3

**Strengths And Weaknesses:**

Strengths:
- Theoretical Foundation: The paper provides a solid mathematical framework grounded in latent concept theory and Bayesian inference. The formalization of the debate process through latent concepts is well-motivated, and the theoretical proofs (Theorems 4.1 and 4.2) establish clear advantages over static aggregation methods under reasonable assumptions.
- Comprehensive Experimental Validation: The evaluation spans multiple domains (hallucination detection, alignment evaluation, reasoning, multi-modal tasks), various model architectures (both proprietary and open-source), and includes thorough ablation studies examining ensemble size, temperature effects, and intervention strategies.
- Practical Relevance: The framework addresses a real need in LLM evaluation systems where single judges can be unreliable and simple aggregation methods fail in complex scenarios.

Weaknesses:
- There are no baseline comparisons with other multi-agent debate frameworks from the past, despite the authors claiming that their method shows significant improvements in judgment accuracy over those that use majority voting.
- The accuracy improvements are often modest (1-3 percentage points). For some datasets like BIG-Bench and JudgeAnything, majority voting sometimes matches or slightly exceeds debate performance, questioning the universal applicability.
- Despite adaptive stopping, the framework still requires multiple model calls and iterative processing, making it computationally expensive compared to single-judge approaches. The scalability to very large ensembles or real-time applications remains unclear.
- There are limited experimental analysis on the experiment results. The paper is essentially just reporting final accuracy numbers and what is shown in the figures without going beyond and providing meaningful interpretations.

---

> ### Author Rebuttal · Authors · 2025-07-31
>
> We sincerely thank Reviewer 1eiT for the thoughtful evaluation and constructive feedback. We greatly appreciate the recognition of our paper's strengths, particularly the solid mathematical framework, the cross-domain experiment validation, and the relevance of addressing the unreliability of a single judge. Below, we address the key concerns and suggestions:
>
> ---
>
> **Weakness 1 - Missing baselines against prior multi-agent debate frameworks**
>
> (1) Positioning of our baselines.
> We treat SoM (simple majority voting) [1] as the baseline for multi-LLM aggregation. Many “debate” style systems either (a) modify majority voting (e.g., RECONCILE’s confidence-weighted voting) [2] or (b) adopt a different interaction protocol (e.g., MAD: two opposing sides + a judge) [3]. These variants lie outside the scope of our framework, which specifically models an iterative, belief-refinement process. Thus, we did not initially position them as direct comparators.
>
> (2) We agree more empirical baselines are needed.
> We fully agree with the reviewer’s suggestion that more empirical baselines are needed.
> Due to time and computation constraint, we conduct experiment using Gemini-2.0-Flash on MAD framework[2], a representative sequential querying approach that structures debates adversarially with multiple debaters presenting arguments for and against a position, moderated by a judge to reach a final decision. The results are somewhat puzzling, as MAD does not even exceed the single-model baseline in accuracy. Our suspicion is that MAD's balanced exposure to both sides gives the incorrect side an equal chance to persuade the judge, skewing outcomes in judgment tasks where nuanced refinement is key.
>
>
> | Dataset           | Single         | SoM            | Debate         | MAD (%)      |
> |-------------------|----------------|----------------|----------------|--------------|
> | **JudgeBench**    | 63.66±3.03     | 66.13±3.72     | **68.06±3.66** | 60.65±1.96   |
> | **LLMBar**        | 76.68±1.97     | 77.75±2.35     | **81.83±2.18** | 73.92±1.27   |
> | **TruthfulQA**    | 69.49±2.71     | 72.01±3.10     | **74.30±2.99** | 70.87±1.59   |
> | **MLLM-Judge**    | 67.50±2.88     | 68.00±3.23     | **69.25±3.19** | 64.20±1.52   |
> | **JudgeAnything** | 81.63±5.70     | 83.46±6.30     | **85.71±5.95** | 71.67±3.36   |
>
> ---
>
> **Weakness 2 - Modest Accuracy Gains and Majority Voting Performance**
>
> We appreciate your observation regarding the modest accuracy gains in Table 1 and cases where majority voting (SoM) matches or slightly exceeds debate performance on BIG-Bench and JudgeAnything. These gains are modest because our framework's iterative refinement adds most value in complex tasks with high initial variance, where collaborative belief updates correct biases (Theorem 4.1), yielding significant improvements. On simpler tasks with high initial consensus, SoM performs comparably or better as refinement introduces minimal benefit, aligning with diminishing returns in low-variance scenarios.
>
> We view this as reasonable and expected, supporting targeted applicability rather than universality: debate excels where accuracy justifies costs, while SoM suffices for straightforward tasks. We will revise Sec. 6 to include statistical significance and task-dependent guidance in the camera-ready version.
>
> ---
>
> **Weakness 3 - Computational Expense and Scalability**
>
> We acknowledge the concern regarding the computational cost of the debate framework. Despite the adaptive stopping mechanism, the iterative nature of the debate process requires multiple model calls, making it more expensive than single-judge approaches. For an ensemble size of 7 (our recommended setting, Appendix B.2), the initial round incurs 7 calls (one per agent), and adaptive stopping typically adds 1–6 more rounds (average ~5, Table 2), resulting in approximately 35 calls total—roughly 35 times the cost of a single-judge query in terms of generations.
>
> Larger ensembles (>11 agents) exacerbate costs due to context dilution and diminishing returns (-0.7% accuracy average, Appendix Table A.1). For real-time applications, the framework suits offline batch processing in high-stakes scenarios (e.g., fact-checking), where accuracy gains (+4.08% on LLMBar) justify overhead. We recommend size 5–7 and will incorporate this clarification, including relative cost multiples, in Sec. 6 of the camera-ready version.
>
> ---
>
> **Weakness 4**: Analytical Limitation.
>
> We recognize the limited in-depth analysis beyond accuracy numbers and figures. Key insights include: bimodal convergence reflecting unanimous alignment or failure (Fig. 2, supporting Theorem 4.1); KS dynamics showing rapid stabilization (Fig. 3); size 7 optimizing accuracy-cost balance (Table B.1); temperature 1.0 maximizing performance (Table B.2); pruning aiding diversity without consistent gains (Table B.3). We will reorganize Appendix B.3 analyses into the main body in the camera-ready version, adding qualitative case studies.
>
> ---
>
> **Question 1**: Conditions for Preferring the Debate Framework Over Simpler Approaches
>
> The debate framework should be preferred over simpler methods like majority voting under the following conditions:
>
> - **Complexity of the Task**: For tasks requiring high levels of reasoning, such as fact-checking or nuanced decision-making (e.g., LLMBar), the debate framework significantly outperforms simpler methods. It allows for the iterative refinement of outputs, leading to higher accuracy.
> - **Task Sensitivity**: In tasks where minor errors have significant consequences (e.g., content moderation or judgment evaluation), the collaborative reasoning process of debate offers better outcomes, justifying the computational overhead. For example, Gemini-2.0-Flash showed substantial improvements in accuracy in high-stakes tasks.
>
> We believe the computational cost is justified when high accuracy is essential, particularly for complex or high-stakes tasks.
>
> ---
>
> **Question 2: The Effect of Ensemble Size**
>
> Our analysis (Appendix B.2, L814) shows that an ensemble size of 7 provides the best balance between accuracy and computational cost across most tasks. Larger ensembles (Size-9 or greater) show diminishing returns in accuracy, while significantly increasing computational costs, smaller ensembles (Size-5) are sufficient to maintain accuracy with minimal cost. We recommend Size-7 as the optimal choice for most use cases.
>
> We will explicitly include this guidance in the main body of the paper.
>
> ---
>
> **Question 3**: Failure Cases
>
> 1. **Small Models**: When using smaller models, the **limited diversity in reasoning** can reduce the effectiveness of iterative refinement in the debate process, making the framework less beneficial.
>
> 2. **Simple Tasks**: For tasks that are **straightforward** or **well-defined**, where the correct answer is obvious, the additional computational cost of the debate framework does not lead to significant improvements. In these cases, **majority voting** can be more efficient.
>
> In such scenarios, the added complexity of the debate framework does not justify the computational overhead. We will update the paper to emphasize these cases and provide clearer guidance on when the debate framework may not offer advantages.
>
> ---
>
> **Reference**
> [1] Yilun Du, Shuang Li, Antonio Torralba, Joshua B. Tenenbaum, and Igor Mordatch. Improving factuality and reasoning in language models through multiagent debate. In Proceedings of the 41st International Conference on Machine Learning, ICML’24. JMLR.org, 2024.
>
> [2] Justin Chen, Swarnadeep Saha, and Mohit Bansal. 2024. ReConcile: Round-Table Conference Improves Reasoning via Consensus among Diverse LLMs. In Proceedings of the 62nd Annual Meeting of the Association for Computational Linguistics (Volume 1: Long Papers), pages 7066–7085, Bangkok, Thailand. Association for Computational Linguistics.
>
> [3] Tian Liang, Zhiwei He, Wenxiang Jiao, Xing Wang, Yan Wang, Rui Wang, Yujiu Yang, Shuming Shi, and Zhaopeng Tu. 2024. Encouraging Divergent Thinking in Large Language Models through Multi-Agent Debate. In Proceedings of the 2024 Conference on Empirical Methods in Natural Language Processing, pages 17889–17904, Miami, Florida, USA. Association for Computational Linguistics.

---

> > ### Comment · Reviewer_1eiT · 2025-08-05
> >
> > Thanks for the author's response, I would like to respectfully keep my score. I believe the mathematical framework is intriguing and meaningful, while I believe the clear comparison with prior empirical works would substantiate the contribution of the paper.

---

> > > ### Author Response · Authors · 2025-08-05
> > >
> > > Thank you once again for your thoughtful review and for engaging with our rebuttal. In the camera-ready version, we will broaden the baseline comparisons by adding further debate-framework variants, with a particular focus on CAMEL.

---

### Note · Authors · 2025-08-14

Dear Reviewers, AC, SAC, and PC,

We sincerely appreciate the constructive feedback provided by the reviewers. Your insightful comments have helped us enhance the quality of our work.

We acknowledge that several reviewers raised concerns about the lack of baseline comparisons in our paper. The reason we only use the majority vote and single answer as baselines is that, in our mathematical framework, only this two can be proven to improve the performance. However, we recognize the importance of empirical comparisons and have therefore conducted additional experiments using the MAD framework [1] and plan to incorporate CAMEL [2] as another baseline for comparison.

Furthermore, we will clarify the specific scenarios in which our framework justifies the additional computational cost, to provide a more comprehensive understanding of its practical applicability.

Thank you again for your valuable feedback.

[1] Tian Liang, Zhiwei He, Wenxiang Jiao, Xing Wang, Yan Wang, Rui Wang, Yujiu Yang, Shuming Shi, and Zhaopeng Tu. 2024. Encouraging Divergent Thinking in Large Language Models through Multi-Agent Debate. In Proceedings of the 2024 Conference on Empirical Methods in Natural Language Processing, pages 17889–17904, Miami, Florida, USA. Association for Computational Linguistics.
[2] CAMEL: Communicative Agents for “Mind” Exploration of Large Language Model Society, NeurIPS 2023

---

### Decision · Program_Chairs · 2025-09-17

**Decision:**

Accept (poster)

**Comment:**

This paper proposes a multi-agent debate framework in which multiple LLMs iteratively refine their judgments through structured discussion, addressing the limitations of static ensemble methods like majority voting. The framework is mathematically formalized using latent concepts, with proofs showing theoretical advantages over static aggregation. To mitigate the computational cost of prolonged debates, the authors introduce an adaptive stability detection mechanism based on a time-varying Beta-Binomial mixture model with Kolmogorov–Smirnov testing, enabling early stopping once consensus emerges. Experiments across diverse benchmarks—including BIG-Bench, TruthfulQA, LLMBar, and six datasets spanning both language and vision tasks—demonstrate that the debate framework consistently outperforms majority voting in accuracy and robustness while maintaining computational efficiency across various model backbones.

The reviewers agreed that the proposed algorithm is novel and enjoys theoretical rigor. The proposal also addresses a practical question in building a multi-turn multi-agent debate system. The paper presents its details clearly and is easy for the audience to follow and reproduce.

Though there were a couple of remaining concerns on the imposed assumptions and a lack of complete empirical comparisons and analysis, I believe this work contributes to the understanding of the working mechanisms of a mult-agent debate system.